

# An Innovative Hybrid SG-CEEMDAN-ARIMA-LSTM Model for Forecasting Meteorological Drought: Trends and Forecasting

Siphamandla Sibiya[1,3], Nkanyiso Mbatha[2], Shaun Ramroop[1], Sileshi Melesse[1]

[1]School of Mathematics, Statistics, and Computer Science, University of KwaZulu-Natal, Pietermaritzburg Campus, Private Bag X01, Scottsville 3209, South Africa.

[2]Council for Scientific and Industrial Research, Holistic Climate Change, Smart Places, Mering Naude Road, Brummeria, Pretoria 0001, South Africa.

[3]Department of Mathematical Sciences, University of Zululand, Private Bag X1001, KwaDlangezwa 3886, South Africa.

*Correspondence to*: Siphamandla Sibiya (Siphasibiya@gmail.com)

**Abstract.** Droughts are defined as extended periods of below-average rainfall resulting in a shortage of water and have significant impacts on ecosystems, agriculture, and water supplies. One of the most challenging aspects of addressing drought is trend patterns and developing accurate prediction models that will be crucial for efficient mitigation and resource management. Analyzing drought is inherently uncertain and complex due to the dynamic and evolving character of climate trends. This study used a special method called the modified Mann-Kendall (MMK) approach and a new trend analysis (ITA) to find trends and introduced a better way to make predictions by using the Standardized Precipitation Index (SPI) along with a combined model that takes advantage of the Savitzky-Golay filter and Complete Ensemble Empirical Mode Decomposition with Adaptive Noise (SG-CEEMDAN) for preparation, plus Autoregressive Integrated Moving Average (ARIMA) and Long Short-Term Memory (LSTM) techniques. In terms of trend analysis of the SPIs, MK and MMK tests revealed a most statistically significant decreasing trend. For example, Pongolapoort Dam showed negative Z-score (p-values) for the SPI-6, SPI-9, and SPI-12 in the MK and MMK tests, which are represented as ($-7.19$ ($6.12e^{-13}$), $-8.74 (< 0.00)$, $-9.83 (< 0.00)$ and $-8.22$ ($2.22e^{-16}$), $-5.44$ ($5.40e^{-8}$), $-6.51$ ($7.41e^{-11}$), respectively. Additionally, the ITA confirmed a significant downward trend across all time scales of the SPI. The SPI forecasting results show that the hybrid model, called SG-CEEMDAN-ARIMA-LSTM, had the best prediction accuracy compared to all other models for every SPI time scale. The coefficient of determination ($R^2$) values of the proposed hybrid model was notably high: 0.9839 for SPI-6, 0.9892 for SPI-9, and 0.9990 for SPI-12. This demonstrates that the hybrid model offers the best fit to the data and is the most suitable choice for forecasting short-to-long-term drought conditions in the uMkhanyakude district. Furthermore, the inclusion of decomposition techniques, such as SG, CEEMDAN, and SG-CEEMDAN, significantly enhances the performance of the model.



## 1. Introduction

Drought is an extended hydrometeorological phenomenon that affects large areas and inflicts considerable harm (Bagmar and Khudri, 2021; Kalisa et al., 2021; Song and Park, 2023). Drought is among the most devastating natural phenomena due to its extensive and enduring impacts (Marengo et al., 2017; Jahanzaib et al., 2021; Dikshit et al., 2022; Gavahi et al., 2022). Drought emerged from climate change, resulting in elevated global temperatures and less precipitation (Chen et al., 2024). Prolonged exceptionally low precipitation leads to water scarcity and can cause soil moisture deficiency, hence significantly jeopardizing

food security (Wang et al., 2023). Drought, a frequent and intricate climatic occurrence, has historically had adverse consequences on economic, environmental, and social sectors, negatively impacting civilization (Hinge et al., 2022; Xu et al., 2022). Drought is defined as extended water deficiency that adversely affects edaphic, hydrological, meteorological, and social elements (Wilhite and Glantz, 1985; Cunha et al., 2019; Adeola et al., 2021; Shang et al., 2023). The occurrence of droughts is acknowledged as a critical factor in water resources planning and management (Bagmar and Khudri, 2021). Drought is a

physical phenomenon and environmental hazard that can lead to catastrophe if reaction skills and vulnerability are insufficient to alleviate its effects (Ruwanza et al., 2022).

In response to global apprehension regarding climate change and its impact on local climates, there has been a heightened focus on analyzing trends in severe drought occurrences in recent years. Several studies have made significant contributions to this area of research, such as the investigations conducted by Labudová et al. (2017), Shiru et al. (2018), Polong et al. (2019),

Danandeh Mehr et al. (2019), Katipoğlu et al. (2022), Phuong et al. (2022), and Gond et al. (2023). Abarghouei et al. (2011) conducted a study in Iran to analyze the linear trends of Standardized Precipitation Index (SPI) indices at 3, 6, 9, 12, and 18-month intervals over a span of 30 years. The study determined that the level of drought in the country is escalating. Caloiero et al. (2018) utilized the Mann-Kendall and ISM techniques to examine potential developments in SPI across New Zealand. They identified both upward and downward trends in various places. In a similar manner, Jin et al. (2020) employed the Mann-

Kendall and ISM techniques to analyze the trends in SPEI series at Zoige Wetland, China. Their findings indicated a prevalent pattern of drying in most of the studied regions. Mehr and Vaheddoost (2020) conducted a study in Ankara Province, Turkey, using the well-known Spearman rank-order correlation coefficient, Sen's method, and trend analysis to look at the trends in SPI and SPEI over 3-, 6-, and 12-month periods. The findings revealed that the province had five instances of severe drought between 1971 and 2016. There were some discrepancies in the timing of these drought occurrences when comparing the

Standardized Precipitation Index (SPI) and the Standardized Precipitation Evapotranspiration Index (SPEI), especially when considering the 6- and 12-month periods. Although the observed drought events displayed a modest decrease in SPEI, the SPI did not demonstrate the same trend. Di Nunno et al. (2024) utilized the seasonal Mann-Kendall test and the Bayesian Changepoint Detection and Time Series Decomposition algorithm to assess the general trends for each cluster and SPI time scale. Additionally, they used these methods to detect sudden changes in trend and seasonality along the SPI time series,

respectively. The findings of the seasonal Mann-Kendall (MK) test indicated statistically significant upward trends in the SPI for all clusters, except for the southeastern region of the United Kingdom, where downward trends were seen, although they



were not statistically significant. Following the abovementioned trend analysis in drought studies, drought forecasts are important for enhancing the understanding of drought dynamics. Drought forecasting plays a key role in providing early drought warnings to mitigate its impacts and improve drought management (Balti et al., 2020; Zhang et al., 2022; Tan et al.,

2024; Zhang et al., 2024).

New and combined/hybrid models have become the main way to predict droughts, effectively handling the complex and changing nature of rainfall data. Consequently, hybrid models are extensively employed in drought research to enhance predictive accuracy. Alquraish et al. (2021) investigated the efficacy of three hybrid models: the Hidden Markov Model-Genetic Algorithm (HMM-GA), ARIMA-Genetic Algorithm (ARIMA-GA), and ARIMA-Genetic Algorithm-Artificial

Neural Network (ARIMA-GA-ANN), comparing them against the benchmarks of HMM and ARIMA for forecasting the Standardized Precipitation Index (SPI) in the Arabian Peninsula. The findings indicated that the hybrid models outperformed the benchmarked standalone model. Ding et al. (2022) employed a hybrid model integrating complementary ensemble empirical mode decomposition (CEEMD) and long short-term memory (LSTM) for predicting the standardized precipitation index (SPI) over various timescales in the Xinjiang Uygur Autonomous Region of China. The findings indicate that as the

timescale increases, the predictive accuracy of the LSTM and CEEMD-LSTM models progressively enhances. Xu et al. (2022) employed a hybrid model integrating CEEMD and ARIMA for the prediction of SPI across various timescales in the Ningxia Hui Autonomous Region. The results indicate that the CEEMD–ARIMA model exhibited strong concordance with the SPI values, demonstrating that the combined model surpassed the performance of the individual model. Latifoglu and Ozger (2023) created a new method that combines phase transfer entropy (pTE) with the Tunable Q Factor Wavelet Transform (TQWT),

which is improved using the Grey Wolf Optimization (GWO) algorithm. The subband data obtained from the SPI are assessed using Artificial Neural Networks (ANN), Support Vector Regression (SVR), Maximum Likelihood (ML), and the Gaussian Process Regression Model (GPR). The findings illustrate the enhanced efficacy of the pTE-GWO-TQWT-ML models compared to alternative approaches. The pTE-GWO-TQWT-GPR model exhibits superior predictive performance compared to the pTE-GWO-TQWT-ANN and pTE-GWO-TQWT-SVR models. Sibiya et al. (2024) used a combined model that merges

CEEMDAN with ARIMA, LSTM, and ARIMA-LSTM methods to predict the Standardized Precipitation Index (SPI) at various time periods (SPI-6, SPI-9, and SPI-12) for Cape Town International Airport. The results indicated a significant alignment between the CEEMDAN-ARIMA-LSTM model and the SPI values, implying that the suggested hybrid model outperformed all other models. The aforementioned studies demonstrated the efficacy and adaptability of tailored hybrid models to meet diverse research objectives and enhance forecasting performance. A researcher must comprehend the strengths

and weaknesses of each model to develop a viable hybrid-based model (Tan et al. 2023). Therefore, using hybrid-based models is suggested because they have become a popular choice for researchers looking for new ways to create very reliable drought forecasting models.

Numerous researchers have developed extensive drought prediction models for the purpose of monitoring drought conditions.

These models are associated with some challenges when applying preprocessing techniques in conjunction with predictive



models (Sibiya et al. 2024). Methods like decomposition techniques (such as EMD, EEMD, CEEMD, CEEMDAN, and VMD) and wavelet transformation require a lot of computer power, especially when working with large amounts of data or in real-time situations. To address these challenges, this study employed the Savitzky-Golay Filter in combination with CEEMDAN for data preprocessing. CEEMDAN can be easily affected by noise and often creates misleading Intrinsic Mode Functions

(IMFs) when used directly on raw data, particularly in unstable time series like drought indices. By smoothing the data first with the Savitzky-Golay Filter, we reduce unnecessary high-frequency noise, which helps CEEMDAN create fewer IMFs and need fewer repetitions. This approach results in a reduced computational burden and yields better-separated intrinsic components. The application of the Savitzky-Golay filter enhances signal quality prior to decomposition, ensuring that meaningful components are retained. A smoother input resulting from Savitzky-Golay filtering facilitates quicker convergence,

subsequently decreasing runtime and memory usage. With reduced noise and clearer IMFs, ARIMA and LSTM models are trained on more informative and stable inputs. This reduces the chance of the models getting confused by noisy data and helps them better understand real patterns in drought behavior, which leads to stronger and more reliable forecasting results. This paper discusses the novel hybrid model used for drought prediction.

The current scenario entails trend analysis to evaluate the trend pattern prior to forecasting the SPI series. The modified Mann-

Kendall method and advanced trend analysis were utilized to examine the drought in the uMkhanyakude district. We identified change points in the SPI time series at intervals of 6, 9, and 12 months. We subsequently examined the SPI trend using a predictive modelling approach. The suggested methodology integrates the advantages of the Savitzky-Golay filter and Complete Ensemble Empirical Mode Decomposition with Adaptive Noise (SG-CEEMDAN) alongside Autoregressive Integrated Moving Average (ARIMA) and Long Short-Term Memory (LSTM) models for the prediction of drought

occurrences. Using the SPI index trend analysis and prediction together is a combined approach that enhances our understanding of how droughts work. We selected the SPI as the drought index due to its simplicity and adaptability. It solely necessitates precipitation data for calculation, rendering it attainable in areas with scarce meteorological information. Its adaptability to diverse temporal scales facilitates a thorough evaluation of drought severity across varying durations. Nonetheless, the SPI has constraints as it relies exclusively on precipitation and neglects variables such as temperature and

soil moisture, which are incorporated in more sophisticated indices like SPEI.

To the authors' knowledge, no prior research has investigated the changes in drought patterns related to trends and forecasted droughts using the predictive hybrid model SG-CEEMDAN-ARIMA-LSTM. Consequently, this model has not been introduced in the literature yet, even for alternative hydrometeorological applications. This paper enhances the evolving domain of climate modeling and drought prediction by introducing a novel hybrid forecasting method, clarifying its constraints,

and suggesting potential directions for future improvements. The results derived from this study were meticulously analysed to provide valuable insights for: (1) the management of water resources to facilitate planning for water allocation and the implementation of adaptive strategies to alleviate drought impacts; (2) the design of infrastructure projects, including





reservoirs, irrigation systems, and water distribution networks, to accommodate evolving hydrological patterns; (3) the establishment of early warning systems for authorities to notify affected regions, thereby enabling prompt responses and
preventive measures; (4) the enhancement of knowledge through the development of hybrid models for forecasting SPI-6, SPI-9, and SPI-12; and (5) the identification of optimal models for meteorological drought forecasting in semi-arid regions.**2.**

**Material Methods**

**2.1. The Study Area**

This study employed monthly precipitation records from 1980 to 2023, obtained from the South African Weather Service
(SAWS) for the uMkhanyakude District in South Africa. The uMkhanyakude District Municipality is located in the far northern region of the KwaZulu-Natal (KZN) province (coordinates: 32.014489° S, 27.622242° E). The municipality covers a total area of 13,855 km², making it the second largest in the province, exceeded only by the Zululand Municipality. The uMkhanyakude District was formed immediately after the local government elections in December 2000 as part of municipal demarcation, encompassing some of the most destitute and underdeveloped areas of KwaZulu-Natal. The uMkhanyakude District consists
of four local municipalities: uMhlabuyalingana, Jozini, Big Five Hlabisa, and Mtubatuba. The municipality is geographically surrounded by Mozambique to the north, the Indian Ocean to the east, the uThungulu River to the south, Zululand to the west, and the Kingdom of Swaziland to the northwest. Figure 1 depicts the spatial arrangement of the stations, superimposed on the National Aeronautics and Space Administration (NASA) Shuttle Radar Topography Mission (SRTM) digital elevation model of the research area (Farr et al., 2007). The stations are situated in a relatively low-lying region in the eastern section of South
Africa.





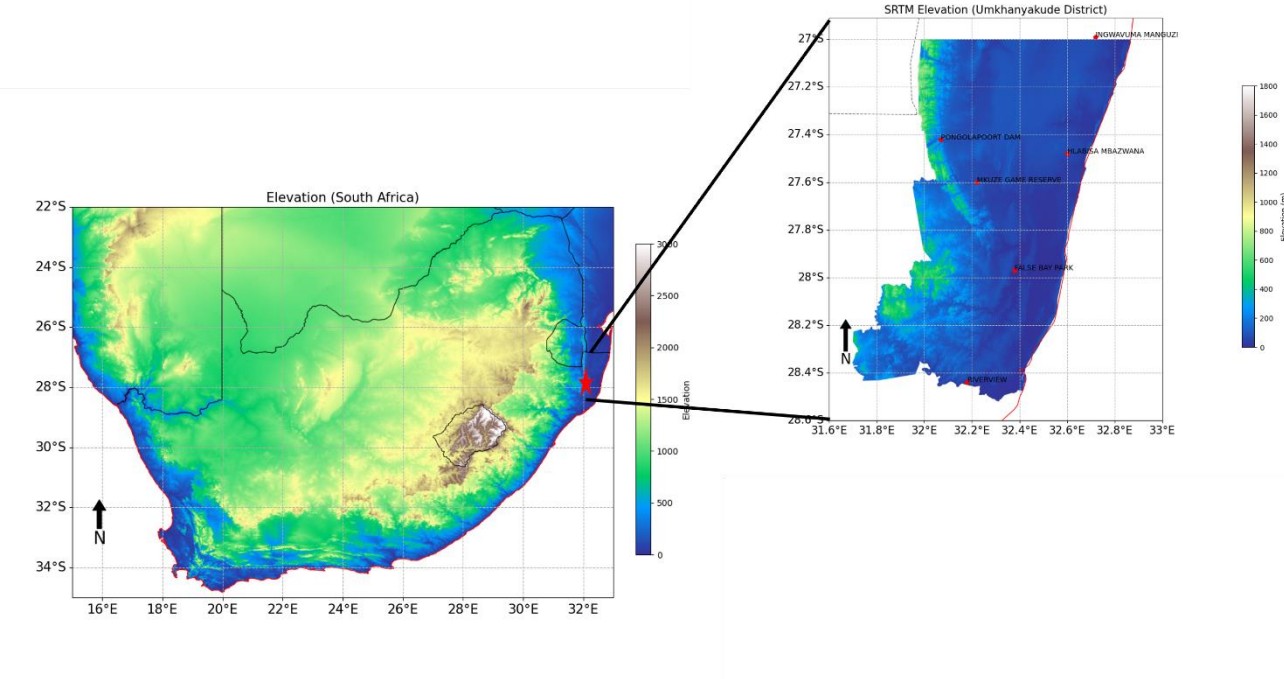

**Figure 1: Overview of the uMkhanyakude District (meteorological stations).**

## 2.2. Modified Mann-Kendall

The modified Mann-Kendall methodology derives from the nonparametric Mann-Kendall method (Mann, 1945; Kendall, 1975), which is extensively employed to detect trends in hydro-meteorological time series (Caloiero et al., 2011; Bard et al., 2015; Wang et al., 2017; Mirabbasi et al., 2020). The modified Mann–Kendall (MMK) test was employed for serially correlated data exhibiting a substantial lag-1 autocorrelation coefficient, utilising the variance correction method proposed by Yue et al. (2002). Hamed and Rao (1998) created this methodology to eradicate all substantial autocorrelation in the time series. Under the assumption that the data are independent and identically distributed, the S statistic of the Mann-Kendall test is computed as follows (Sharifi et al. 2024):

$$S = \sum_{i=1}^{n-1} \sum_{j=i+1}^{n} Sign(x_j - x_i) \tag{1}$$



where n denote the sample size; $x_i$ and $x_j$ denote sequential $ith$ and $jth$ data points, respectively, and sign(.) is the sign function which can be computed as

$$Sign(x_j - x_i) = \begin{cases} 1, & if\ x_j - x_i > 0 \\ 0, & if\ x_j - x_i = 0 \\ -1, & if\ x_j - x_i < 0 \end{cases} \tag{2}$$

with the mean and variance of the $S$ statistics in equation are as follows (Helsel and Hirsch 1993; Ma et al. 2014; Ashraf et al. 2023)

$$E(S) = 0 \tag{3}$$

$$Var(S) = \frac{n(n-1)(2n+5) - \sum_{i=1}^{p} t_i(t_i - 1)(2t_i + 5)}{18} \tag{4}$$


where $p$ is the number of tied groups and $t_i$ denotes the number of data points in the $t^{th}$ group. The second term represents an adjustment for tied group or censored data. The standardized Z statistic is calculated as

$$Z_{MK} = \begin{cases} \dfrac{S-1}{\sqrt{Var(S)}}, & S > 0 \\ 0, & S = 0 \\ \dfrac{S+1}{\sqrt{Var(S)}}, & S < 0 \end{cases} \tag{5}$$

The test statistic Z is used to measure the significance of the trends. In the modified Mann-Kendall approach, a modified variance of S is computed as follows (Hamed and Rao, 1998)

$$Var(S^*) = Var(S) \cdot \frac{n}{n^*} \tag{6}$$

where $n^*$ is the effective sample size. The $\frac{n}{n^*}$ ratio can be calculated as follows (Hamed and Rao, 1998)

$$\frac{n}{n^*} = 1 + \frac{2}{n(n-1)(n-2)} \sum_{i=1}^{n} (n-i)(n-i-1)(n-i-2) r_i \tag{7}$$

where $r_i$ denotes the lag-$i$ significant autocorrelation coefficient of rank $i$ in a time series. Then the standardized statistic of the S statistic, denoted as Z, can be derived as

$$Z_{MMK} = \begin{cases} \dfrac{S-1}{\sqrt{Var(S^*)}}, & S > 0 \\ 0, & S = 0 \\ \dfrac{S+1}{\sqrt{Var(S^*)}}, & S < 0 \end{cases} \tag{8}$$



If the calculated Z values ($Z_{MK}$ and $Z_{MMK}$) exceed the critical values of $-Z_{1-\alpha/2}$ or fall below $Z_{1-\alpha/2}$, there is no discernible trend in the time series at the significance level of α. If the Z value is positive and exceeds $Z_{1-\alpha/2}$, the trend is upward; conversely, if the Z value is negative and falls below $-Z_{1-\alpha/2}$, the trend is downward.

### 2.3. Innovative Trend Analysis

The Innovative Trend Analysis (ITA) method, initially introduced by Sen (2012), has been widely employed for detecting patterns in precipitation time series. Since its debut, the ITA technique has experienced substantial improvements in both mathematical and graphical aspects, as evidenced by Şen (2017) and Alashan (2018). The ITA method does not depend on assumptions of serial autocorrelation, normalcy, or record length, making it appropriate for both graphical and statistical trend analysis (Besha et al., 2022). Initially, the time series is bifurcated into two equal segments and organised in ascending order.

The initial segment of the time series ($x_i : i = 1, 2, \ldots, n/2$) is positioned along the horizontal x-axis, while the subsequent segment ($x_j : j = n/2 + 1, n/2 + 2, \ldots, n$) is situated along the vertical y-axis in the Cartesian coordinate system (Ashraf et al. 2023). The ITA approach visually represents trend analysis, specifically indicating monotonic growing, declining, and trendless circumstances (Oztopal and Şen, 2017; Likinaw et al., 2023). A monotonically growing or declining trend can be identified when the majority of points are situated above or below the 45° (1:1 line), respectively. A trendless condition arises

when the data points are clustered along the 45° line (Şen, 2012). We employ the magnitude of the slope parameter to convey information about monotonicity. The slope parameter of the ITA technique is a stochastic property dependent on the sample means of the first half ($n_1$) and the second half ($n_2$) of the time-series mean data values. According to Şen (2017), the straight-line trend slope ($S_{ITA}$) can be estimated using the following expression:

$$S_{ITA} = \frac{2x(x_j - x_i)}{n} \tag{9}$$

where n represents the total number of observations, $x_i$ and $x_j$ are the arithmetic means of the first and second halves of the

sub-series, respectively. Given that $x_i$ and $x_j$ are stochastic variables, the expected value of the slope can be determined by analysing the expectancies of both the first and second halves of the time series (Alashan, 2020; Harka et al., 2021):

$$E(S_{ITA}) = \frac{2}{n}\left[E(x_j) - E(x_i)\right] \tag{10}$$

For the no trend condition, $E(x_j) = E(x_i)$, the $E(S_{ITA}) = 0$ and standard deviation (SD) of the two half time-series $\left(\sigma_{x_j} = \sigma_{x_i} = \sigma/\sqrt{n}\right)$, σ is the SD is of the parent series. If $E(x_j) \neq E(x_i)$, the differences between $E(x_j)$ and $E(x_i)$ gives the variance

$$\sigma_{S_{ITA}}^2 = \frac{8}{n^2}\left[E(x_j) - E(x_j x_i)\right] \tag{11}$$

and the SD of the slope



$$\sigma_{S_{ITA}} = \frac{2\sqrt{2}}{n\sqrt{n}} \sigma \sqrt{1 - \rho_{x_j x_i}} \tag{12}$$

In the stochastic processes, the term $\rho_{x_j x_i}$ is the correlation coefficient between the two mean values, and can be estimated as

$$\rho_{x_j x_i} = \frac{E(x_j x_i) - E(x_j)E(x_i)}{\sigma_{x_j} \sigma_{x_i}} \tag{13}$$

In the end, the upper and lower confidence limit (CL) of the trend slope was calculated (Şen 2017):

$$CL_{(1-\alpha)} = 0 \pm \left(Z_{1-\alpha/2}\right) \sigma_{S_{ITA}} \tag{14}$$

$Z_{1-\alpha/2}$ denotes the crucial slope for standardised time-series at ±1.96 for a 95% significance level or ±1.645 for a 90% significance level (Alashan, 2020). If the ITA slope value is beyond the lower and upper confidence limits, the null hypothesis of no significant trend should be rejected at the α significance level (Şen, 2017). In a two-tailed scenario, the null hypothesis ($H_0$) posits the absence of a trend in time-series data, while the alternative hypothesis ($H_1$) asserts the presence of a trend in time-series data at a significance level of α. If the slope, $\pm S_{ITA} > \pm CL_{(1-\alpha)}$, then ($H_0$) is discarded in favour of ($H_1$). The positive and negative values of $S_{ITA}$ signify an upward and downward trend in the time-series data, respectively (Şen, 2017).

## 2.4. The SPI Calculation

For the purpose of analysing the severity of drought which is caused by a lack of water supply as a result of reduced precipitation in response to rising demand, the SPI was designed by McKee et al. (1993) and is based on probability (Zuo, 2021). Based on the cumulative likelihood of a specific amount of precipitation, the SPI indicator is calculated by fitting the precipitation throughout the same period with a certain distribution function. At its largest point, the SPI index represents the quantile of a normal distribution. Each time axis has an estimated drought index for 6, 9, and 12 months. This is based on the gamma probability density function, which considers the periodic distribution of precipitation for the corresponding data point. The expression of the density function for this distribution is as follow.

$$g(x) = \frac{1}{\beta^\alpha \Gamma(\alpha)} x^{\alpha-1} e^{-\frac{x}{\beta}} \tag{15}$$

where $\alpha$ is the shape parameter, $\beta$ is the scale parameter and $x$ is the precipitation amount, and $\Gamma(\alpha) = \int_0^\infty y^{\alpha-1} e^{-y} dy$ is gamma function. The maximum likelihood estimates of the parameters $\alpha$ and $\beta$ are:

$$\alpha = \frac{1}{4A} \left(1 + \sqrt{1 + \frac{4A}{3}}\right) \tag{16}$$

$$\beta = \frac{\bar{x}}{n} \tag{17}$$





where $A = \ln(\bar{x}) - \frac{\sum \ln(x)}{n}$, $\bar{x}$ is the precipitation average and $n$ is the sample size. The following equation applies the acquired parameters to the cumulative probability distribution:

$$G(x) = \int_0^x g(x)dx = \frac{1}{\beta^\alpha \Gamma(\alpha)} \int_0^x x^{\alpha-1} e^{-\frac{x}{\beta}} dx \tag{18}$$

G(x) represents the likelihood that the precipitation will be equal to or less than x. The distribution function for precipitation needs to be adjusted because the real precipitation samples can contain a value of 0. Based on this, we can calculate the cumulative probability as:

$$H(x) = q + (1-q)G(x) \tag{19}$$

where q denotes the probability when precipitation equals zero. The probability of no rainfall, q, can be articulated as q = m/r, where m represents the number of days without rainfall and r denotes the number of days with rainfall (Song and Park, 2021).

Consequently, H(x) is converted to the conventional random variable Z of the standard normal distribution, characterised by a mean of 0 and a variance of 1, resulting in:

$$SPI = Z = \begin{cases} -\left(k - \dfrac{c_0 + c_1 k + c_2 k^2}{1 + d_1 k + d_2 k^2 + d_3 k^3}\right), & 0 < H(x) \le 0.5 \\ +\left(k - \dfrac{c_0 + c_1 k + c_2 k^2}{1 + d_1 k + d_2 k^2 + d_3 k^3}\right), & 0.5 < H(x) \le 1.0 \end{cases} \tag{20}$$

$$k = \begin{cases} \sqrt{\ln\left(\left(\dfrac{1}{H(x)}\right)^2\right)}, & 0 < H(x) \le 0.5 \\ \sqrt{\ln\left(\left(\dfrac{1}{1-H(x)}\right)^2\right)}, & 0.5 < H(x) \le 1.0 \end{cases} \tag{21}$$

where $c_0 = 2,515517$, $c_1 = 0.802853$, $c_2 = 0,010328$, $d_1 = 1,432788$, $d_2 = 0,189269$, $d_3 = 0,001308$ are constants. Furthermore, the SPI indicator is a standardised normalised index, establishing a correlational relationship with likelihood.

Table 1 presents the probability associated with each category of drought.

**Table 1. Drought classification using SPI values and corresponding event probability (Llyod-Hughes and Sanders 2002).**






| SPI Values | Drought Category | Probability (%) |
|---|---|---|
| $2.00 \leq SPI$ | Extremely wet | 2.3 |
| $1.50 \leq SPI \leq 1.99$ | Severely wet | 4.4 |
| $1.00 \leq SPI \leq 1.49$ | Moderately wet | 9.2 |
| $0.00 \leq SPI \leq 0.99$ | Mildly wet | 34.1 |
| $-0.99 \leq SPI \leq 0.00$ | Mild dry | 34.1 |
| $-1.49 \leq SPI \leq -1.00$ | Moderate dry | 9.2 |
| $-1.99 \leq SPI \leq -1.50$ | Severe dry | 4.4 |
| $SPI \leq -2.00$ | Extreme dry | 2.3 |

**2.5. The Savitzky-Golay Filter**

The Savitzky-Golay (SG) smoothing technique is a prevalent method employed for noise filtration. Savitzky and Golay (1994) introduced the SG filter as an effective technique for signal smoothing. The SG technique attenuates noise utilising two
parameters: polynomial order and window size. By flexibly adjusting these two parameters, the SG filter can achieve exceptional performance in various pre-processing circumstances. The essence of this procedure involves fitting a low-degree polynomial to the samples within a sliding window using the least squares method, resulting in a new smoothed value for the central point derived by convolution. The SG filter is a specific variant of low-pass filter that substitutes each value in the time series with a new value derived from a polynomial fit to $2m + 1$ surrounding points, including the point to be smoothed, where
m is equal to or larger than the polynomial's order. The polynomial is articulated as follows:

$$p(n) = \sum_{k=0}^{N} a_k n^k \tag{22}$$

where $N$ is the power of the polynomial and $N \leq 2M + 1$. The following equation is used to determine the error between the estimated and original values; in order to find the desired polynomial result, this error must be minimised.

$$\epsilon_N = \sum_{n=-M}^{M} (p(n) - x[n])^2 \tag{23}$$

The following form of discrete convolution can be used to express the filter's output:

$$y[n] = \sum_{m=-M}^{M} h[m]\, x[n - m] = \sum_{m=n-M}^{n+M} h[n - m]\, x[m] \tag{24}$$

This work employs the SG filter for two primary reasons: firstly, it enhances system performance by preserving the width and
height of waveform peaks in noisy SPI, and secondly, it modifies the SPI while maintaining its fundamental qualities.



### 2.6. The Complete Ensemble Empirical Mode Decomposition with Adaptive Noise.

The model's ability to fit functions and converge will be constrained by the complexity and volatility of the original time sequence, which in turn limits the model's predictive power. To overcome this challenge, the complete ensemble empirical

mode decomposition (CEEMDAN) technique is used to pre-process the original nonstationary and nonlinear time sequence. Both empirical mode decomposition (EMD) and ensemble empirical mode decomposition (EEMD), have been enhanced by the CEEMDAN. The computational efficiency is improved, and the reconstructed sequences of both the EMD and EEMD algorithms are free of modal confusion and noise residuals (Zhang et al., 2023). A residual term and a sequence of intrinsic mode functions (IMFs) are the building blocks of a complicated time series signal that the CEEMDAN breaks down.

Step 1: Incorporate a constrained quantity of adaptive white noise into the original sequence $x(t)\delta_0\omega^i(t)$ $(t = 1, 2, 3, \cdots, N)$

$$x^i(t) = x(t) + \delta_0\omega^i(t) \tag{25}$$

where N denotes the number of trials, $\delta_0$ signifies a coefficient of intensity, and $\omega^i(t)$ indicates the ith realisation of a stochastic Gaussian process.

Step 2: The residual $r_1(t)$ and the first modal component $IMF_1$ are obtained by decomposing each equation (1) using EMD.

$$\overline{IMF_1(t)} = \frac{1}{N}\sum_{i=1}^{N} EMD_1\left(x^i(t)\right) \tag{26}$$

$$r_1(t) = x(t) - \overline{IMF_1(t)} \tag{27}$$


In this context, $EMD_1(.)$ denotes the initial IMF component produced by the EMD algorithm, while $r_1(t)$ signifies the residual associated with the first stage.

Step 3: Add white noise $\delta_1 EMD_1(\omega^i(t))$ to the residual $r_1(t)$ and further decomposed by EMD to obtain the second modal component $IMF_2$ and residual $r_2(t)$.

$$\overline{IMF_2(t)} = \frac{1}{N}\sum_{i=1}^{N} EMD_1\left(r_1(t) + \delta_1 EMD_1(\omega^i(t))\right) \tag{28}$$


$$r_2(t) = r_1(t) - \overline{IMF_2(t)} \tag{29}$$

For the $j = 3, 4, \cdots, N$, the jth IMF component and the jth residual can be computed as:

$$\overline{IMF_j(t)} = \frac{1}{N}\sum_{i=1}^{N} EMD_1\left(r_{j-1}(t) + \delta_{j-1} EMD_{j-1}(\omega^i(t))\right) \tag{30}$$





$$r_j(t) = r_{j-1}(t) - \overline{IMF_j(t)} \tag{31}$$

where $EMD_{j-1}(.)$ denotes the $(j-1)$th intrinsic mode function component derived from the empirical mode decomposition technique, and $r_j(t)$ represents the residual following the jth decomposition.

Step 3: Continue executing step 3 until the residual $r_j(t)$ meets a predetermined termination criterion.

The time series $x(t)$ can ultimately be articulated as

$$x(t) = \sum_{i=1}^{N} \overline{IMF_N(t)} + r_N(t) \tag{32}$$

**2.7. The Autoregressive Integrated Moving Average Model**

The Autoregressive Integrated Moving Average (ARIMA) model, pioneered by Box and Jenkins in the 1970s, serves as a
robust and effective forecasting approach for time series analysis (Box et al., 2015). The ARIMA model, often known as the Box-Jenkins approach, is depicted through the concepts presented by Sibiya et al. (2024) in Figure 2. The ARIMA models predict future values of the time series as a linear combination of historical and residual data. This model comprises three components: the order of seasonal differentiation, autoregressive order, and moving average order (Montgomery et al., 2015). The backward shift operator B is employed to eliminate nonstationarity. A time series, $y_t$, is called homogeneous nonstationary
if it first order difference, $\omega_y = (1-B)y_t = y_t - y_{t-1}$ or the dth difference $\omega_t = (1-B)^d y_t$ is also stationary time series. Furthermore, $y_t$ is referred to as an ARIMA model with orders $p, d$ and $q$, noted $ARIMA(p, d, q)$. Hence, an $ARIMA(p, d, q)$ is often expressed as

$$\phi(B)(1-B)^d y_t = c + \theta(B)\varepsilon_t \tag{33}$$

$$\phi(B) = 1 - \sum_{i=1}^{p} \phi_i B^i \quad \text{and} \quad \theta(B) = 1 - \sum_{i=1}^{q} \theta_i B^i \tag{34}$$

The backward shift operators for $AR(p)$ and $MA(q)$ are defined as $\phi(B)y_t = c + \varepsilon_t$ and $y_t = \mu + \theta(B)\varepsilon_t$ with $c = \mu - \phi\mu$, where $\mu$ and $\varepsilon_t$ are the mean and white noise, respectively and the $\varepsilon_t$ is independent and normal distributed with mean 0 and variance of $\sigma_\varepsilon^2$.



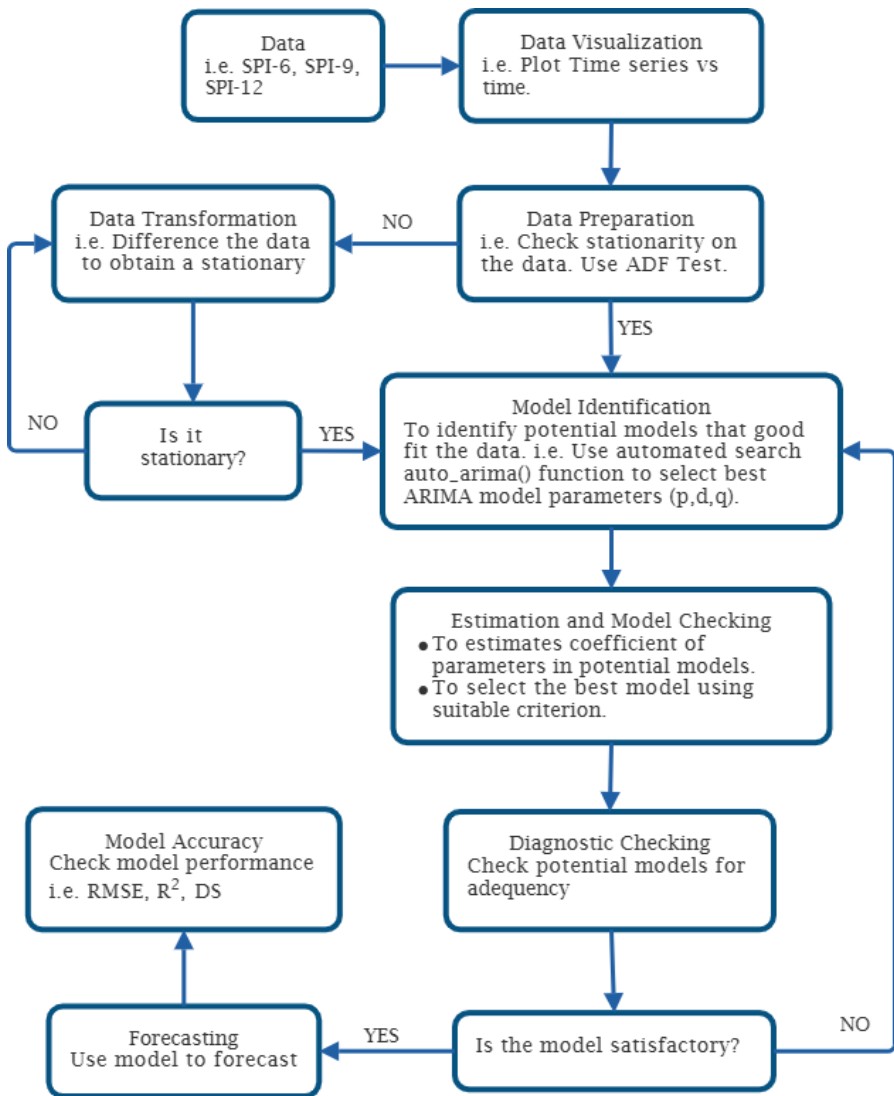

**Figure 2: The Box-Jenkins Steps Approach.**


## 2.8. The Long Short-Term Memory

Long short-term memory (LSTM) algorithms represent a category of recurrent neural network (RNN) designs that are proficient in handling sequential input and identifying temporal relationships (Hochreiter and Schmidhuber, 1997). LSTM networks incorporate specific memory cells and gates for the efficient management and regulation of information flow over

various time steps. Consequently, they can effectively represent the data input while maintaining essential dependencies and patterns. The LSTM methodology addresses the problem of vanishing gradients encountered by RNN algorithms. This occurs when the gradient diminishes to a level insufficient for effectively updating the weights throughout prolonged sequences. The



LSTM facilitates the flow of gradients across time by employing memory cells and gates. The model's foundational design primarily consists of three control gates: input, forget, and output. The activation function is represented by σ, whereas the cell states at time $t - 1$ and $t$ are designated as $C_{t-1}$ and $C_t$ respectively. At time $t$ and time $t - 1$, the cell possesses two concealed states, $h_t$ and $h_{t-1}$. Figure 3 illustrates the building of the LSTM unit, and the mathematical equations (35) to (40) for the LSTM method are provided below. Initially, by employing the model's forget gate, we may determine the current hidden state $h_{t-1}$ and the degree to which the input $x_t$ has been preserved. The formula is

$$f_t = \sigma\left(W_f x_t + U_f h_{t-1} + b_f\right) \tag{35}$$

Secondly, the input gate allows us to ascertain the volume of content from the input variable that can be retained in the cell state $C_t$

$$i_t = \sigma(W_i x_t + U_i h_{t-1} + b_i) \tag{36}$$

$$\tilde{C}_t = \sigma_c(W_c x_t + U_i h_{t-1} + b_i) \tag{37}$$

$$C_t = f_t \odot C_{t-1} + i_i \odot \tilde{C}_t \tag{38}$$

The output gate of the LSTM produces outputs, and the hidden state of each cell is represented by the formula:

$$o_t = \sigma(W_o x_t + U_o h_{t-1} + b_o) \tag{39}$$

$$h_t = O_t \odot \sigma_h(C_t) \tag{40}$$

In the aforementioned formulas, $W_f$, $W_i$, and $W_o$ represent the weight matrices associated with the various control gates. The terms $b_f$, $b_i$, and $b_o$ correspond to the bias terms for each respective control gate. The notation $\tilde{C}_t$ signifies the complete input activation vector, while the operator $\odot$ (Hadamard product) indicates the element-wise multiplication of the elements between two vectors. The $\sigma$ activation function quantifies the amount of information that is transmitted through the various control gates.



Figure 3: Structure diagram of LSTM model.

## 2.9. The ARIMA-LSTM hybrid Model

Achieving accurate estimates of SPI index values through a forecasting model is essential for informed decision-making. Zhang (2003) offers a hybrid model wherein the ARIMA model extracts and predicts linear components, while the residuals, representing nonlinear data subcomponents, are then modelled by the LSTM approach. This study employs a hybrid model that integrates ARIMA and LSTM to predict both linear and nonlinear behaviours with optimal accuracy.

$$\mathcal{H}_t = \mathcal{L}_t + \aleph_t \tag{41}$$

where $\mathcal{L}_t$ and $\aleph_t$ denotes the linear and nonlinear components, respectively, for the hybrid technique which are computed using the initial time series ($y_t$). Consider the original dataset at time t and the forecast results obtained from applying the ARIMA




model as $\hat{\mathcal{L}}_t$ the prediction results. Thus, $\mathcal{E}_t = y_t - \hat{\mathcal{L}}_t$ is the definition of the residual $\mathcal{E}_t$ that is derived by removing $\hat{\mathcal{L}}_t$ from $y_t$. Subsequently we compute the value $\hat{\aleph}_t$ by feeding the series of residuals into the LSTM model, which predicts the nonlinear component of the values. This equation may be written as

$$\hat{\aleph}_t = f_{LSTM}(\mathcal{E}_{t-1}, \mathcal{E}_{t-2}, \dots, \mathcal{E}_{t-n}) + \epsilon_t, \tag{42}$$

340   where is a nonlinear expression associated with the LSTM model and $\epsilon_t$ is the random error. The combined forecasts from the two steps were then used to determine the value for the ARIMA-LSTM hybrid model. As illustrated in Figure 4, the equation $\hat{\mathcal{H}}_t = \hat{\mathcal{L}}_t + \hat{\aleph}_t$ predicts the linearity and nonlinearity values, respectively, using ARIMA and LSTM models.

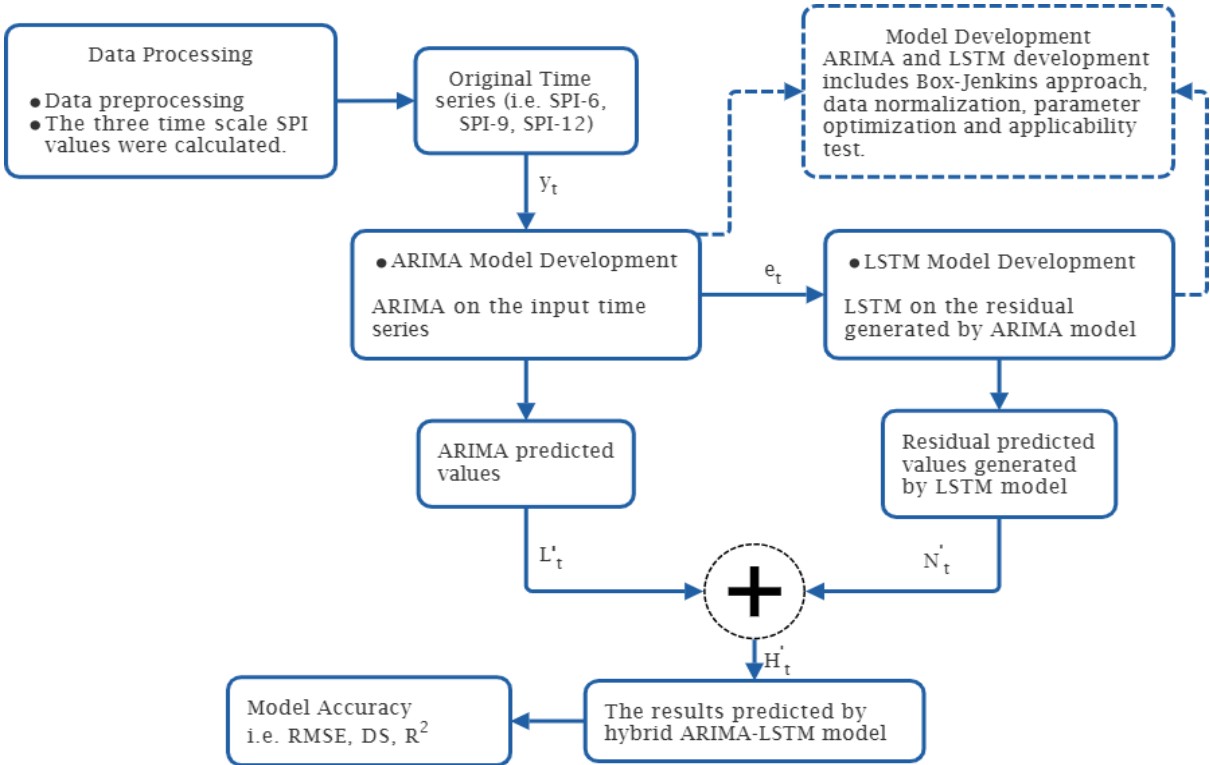

**Figure 4: Predictive flowchart of the ARIMA-LSTM hybrid model.**

345

## 2.10.    The development of the proposed SG-CEEMDAN-ARIMA-LSTM hybrid model

Due to the great uncertainty of the drought data and the existence of complexity, nonlinearity and nonstationary trends, the single prediction model is greatly limited, however the hybrid method has better prediction accuracy. The SG-CEEMDAN-ARIMA-LSTM algorithm that combines different techniques for improved accuracy in predicting drought based on the standardised precipitation index is proposed this study. This hybrid model is designed as a sequential framework where each step refines the data for subsequent modelling. The SG-CEEMDAN pre-processing stage enhances the data by smoothing and





decompose into the meaningful components. The components fed to the ARIMA-LSTM model that involves two-step process: the ARIMA for initial prediction and the LSTM model for refining and enhancing predictions. The hybrid model combines the ARIMA and the LSTM predictions to form the final hybrid forecasts. Figure 5 illustrates the proposed hybrid model algorithm. The process of SPI prediction based on ARIMA-LSTM combined with SG and CEEMDAN as is shown in Figure 5. The process of the data smoothing, decomposition and prediction includes four main steps.

Step 1: Data smoothing and decomposition: SG is first applied to SPI series then CEEMDAN decomposed data to obtain IMFs and residual.

Step 2: A training set and a test are created from sequence that was obtained from step 1.

Step 3: Create the SG-CEEMDAN-ARIMA-LSTM prediction model.

Step 4: Evaluate the prediction model. The SG-CEEMDAN-ARIMA-LSTM model is evaluated by statistical criterion and Taylor diagram.

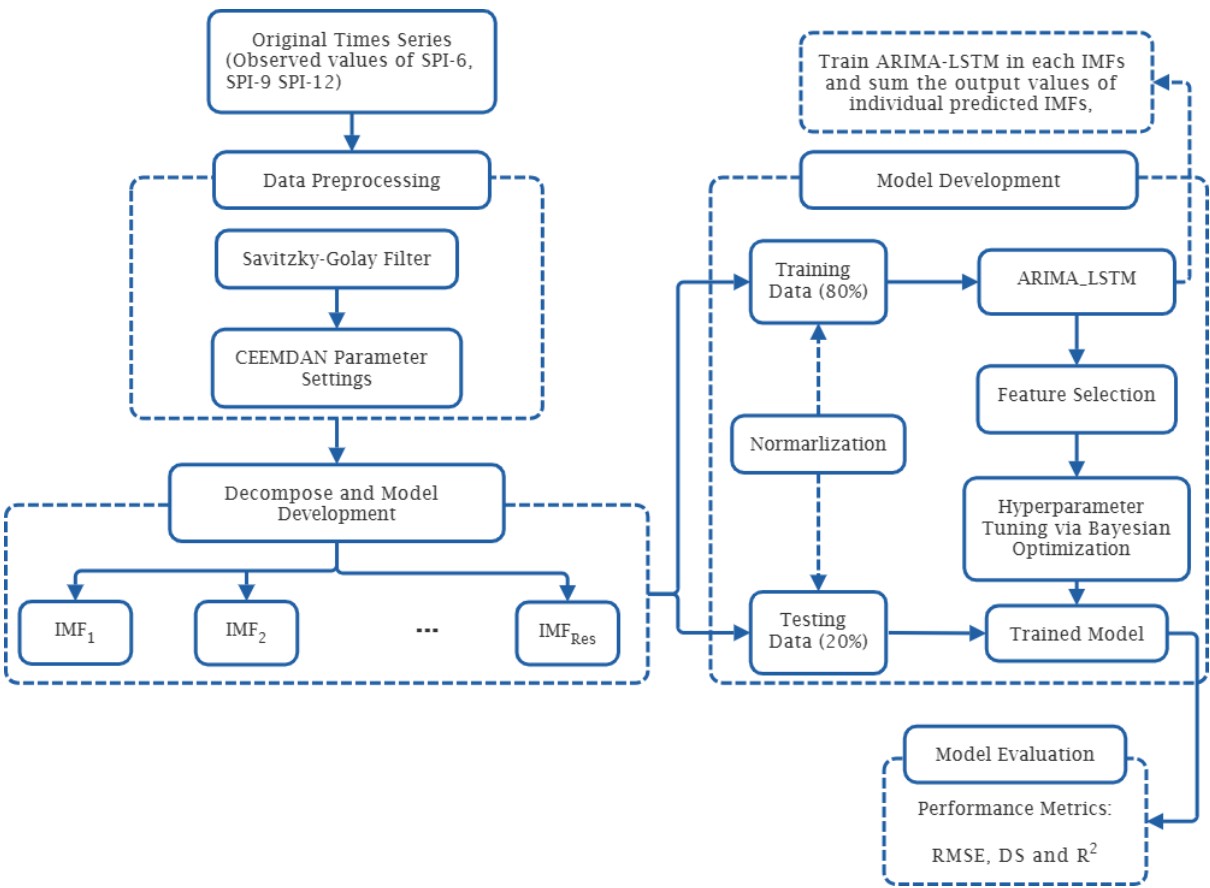

**Figure 5**: **Procedure of proposed SG-CEEMDAN-ARIMA-LSTM hybrid model.**



## 2.11.  Performance Evaluation

To establish the predictive superiority of the SG-CEEMDAN-ARIMA-LSTM model, a comparison was conducted against
other models, including ARIMA, LSTM, ARIMA-LSTM, SG-ARIMA-LSTM, and CEEMDAN-ARIMA-LSTM models. The

performance of the proposed hybrid-based model is evaluated using three indicators namely, root mean square error (RMSE),

coefficient of determination ($R^2$) and directional symmetry (DS). The high value of $R^2$ and DS reflects the better performance

of the forecasting model while the lower the value of RMSE illustrates better forecasting performance.

$$RMSE = \sqrt{\frac{1}{n}\sum_{i=1}^{n}(y_i - \hat{y}_{avg})^2}$$

(43)

$$R^2 = \frac{\left[\sum_{i=1}^{n}(y_i - y_{avg})(\hat{y}_i - \hat{y}_{avg})\right]^2}{\sum_{i=1}^{n}(y_i - y_{avg})^2 \sum_{i=1}^{n}(\hat{y}_i - \hat{y}_{avg})^2}$$

(44)

$$DS = \frac{100}{n-1}\sum_{i=2}^{n}d_i$$

(45)

where

$$d_i = \begin{cases} 1, & (y_i - y_{i-1})(\hat{y}_i - \hat{y}_{i-1}) > 0 \\ 0, & otherwise \end{cases}$$

(46)

$n$ is number of data points, $y_i$ and $\hat{y}_i$ observed and forecasted, respectively. $y_{avg}$ and $\hat{y}_{avg}$ an average of the actual and

forecasted values, respectively. Furthermore, this study conducts a qualitative evaluation of the prediction model's performance

using a Taylor diagram (Taylor, 2001). The Taylor diagram offers a statistical evaluation of the degree of agreement between

the models in terms of their SD, RMSE, and $R^2$, while providing a concise summary of the correspondence between predicted

and observed values. The differences in SD, RMSE, and $R^2$ values among the prediction models are depicted as individual

points on a two-dimensional plot within the Taylor diagram. This diagram, though it follows a common structure, proves

especially valuable when evaluating intricate models.

## 3.   Results and Discussion
### 3.1. Rainfall Data Series

Figure 6 illustrates the daily and monthly cumulative precipitation data recorded at the uMkhanyakude district meteorological

stations in KwaZulu-Natal province, South Africa, from the early 1980s to 2023. The data comprising 20% was employed for

prediction, whereas the data representing 80% was applied for training. The SPI computed utilising rainfall data from





meteorological stations in the uMkhanyakude district, which provide sufficiently extensive records and a consistent structure (Hırca et al., 2022).

**Figure 6: Time series plots of daily and monthly total rainfall data for uMkhanyakude district from early 1980's to 2023. The (left) plot shows the daily rainfall data in millimeters (mm), illustrating the high variability and intermittent nature of daily rainfall events over the years. The (right) plot presents the monthly total rainfall data (mm), which smooths out the daily variability and reveals clearer patterns of rainfall distribution over time.**



3.2. *SPI Time Series and Trend Analysis*

This study SPI values for the 6-, 9-, and 12-month intervals were computed using the monthly mean time series shown in Figure 6. Figure 7 illustrates the time series of the SPI calculated for the 6-month (SPI-6), 9-month (SPI-9), and 12-month (SPI-12) intervals. All SPIs (SPI-6, SPI-9, and SPI-12) demonstrate numerous occurrences of moderate to severe droughts in the studied area. A significant drought episode was reported from late 2004 to 2009. Moreover, SPI-12 demonstrates a

persistent drought spell that began between 2014 and 2016, leading to a decline in water supply conditions in the region (Bukhosini and Moyo, 2023). The statistics across all timelines indicate a troubling trend of extended and intense drought conditions in recent years. This underscores the pressing necessity for efficient water management and drought readiness in the area. Initially, we assess the trend throughout the research area employing nonparametric techniques. The ensuing conclusions will be obtained via advanced trend analysis methods employed to investigate SPI trends.





**Figure 7: Standardized Precipitation Index (SPI) time series plots for uMkhanyakude district over 6-month (SPI-6), 9-month (SPI-9), and 12-month (SPI-12) periods from early 1980's to 2023. Positive SPI values (blue bars) indicate wetter-than-normal conditions, while negative SPI values (red bars) indicate drier-than-normal conditions.**




Figure 8 illustrates the regional outcomes of the ITA methodology used to the 6-, 9-, and 12-month SPI series to ascertain the potential meteorological drought trend in the uMkhanyakude district. Figure 8 includes two vertical bands to elucidate the potential trends of arid and humid conditions: a red band indicating the drought threshold (SPI = -1.5) and a blue band denoting the wet threshold (SPI = 1.5). The zone between the two bands signifies normal conditions, hence facilitating the depiction of both low and high SPI trends using the ITA methodology. Each plot compares the first and second halves of the data series to

identify trends.

In general, for both Figure 8 and Table 3 all stations except Riverview indicate downwards trend for all time scales, in terms of the ITA. For an example, the ITA results obtained using 6-month SPI values exhibit a slightly decreasing trend in precipitation, moving toward the upper right quadrant, indicating the detection of dryer conditions over the 6-month timescale.

Some points approach the severely wet threshold but do not cross it, indicating that there were no extreme wet periods, though some drier periods are evident near the severe dry line. The ITA results obtained using 9-month SPI values shows a more pronounced decreasing trend, indicates a relatively weaker increase in wet conditions over a 9-month timescale. Several points come close to the severe dry threshold, but the data remains mostly within the 95% confidence bounds, indicating moderate variability in precipitation trends. On the other hand, the SPI-12 plot  demonstrates a noticeable decreasing trend toward

dryness, as many points fall below the no-trend line and approach the severe dry region. Riverview indicates the increasing trend across all time scales. The increasing distance between the black dots and the no-trend line highlights a shift toward drier conditions in the second half of the series. In general, the analysis suggests a gradual increase in precipitation for shorter periods (SPI-6), moderate upward trends for medium-term periods (SPI-9), and a more substantial shift toward dry conditions over longer periods (SPI-12) for Riverview. The variability is evident, but there is a clear progression toward drier conditions,

particularly in the SPI-12 plot. This observation could be indicative of changing precipitation patterns, which is crucial for understanding drought risk and informing water resource management strategies.



**Figure 8: Results of Innovative trend analysis applied to different time scales values (SPI-6 (left), SPI-9 (middle), SPI-12 (right)). The blue shaded area represents the 95% confidence level area. The red and blue vertical lines represent the severe drought and severely wet, respectively.**





Table 2 presents the results of the Mann-Kendall (MK) and Modified Mann-Kendall (MMK) trend tests for the Standardized Precipitation Index (SPI) over 6-month (SPI-6), 9-month (SPI-9), and 12-month (SPI-12) periods. The results indicate that across five stations all time scales both MK and MMK methods showed significant decreasing trend with negative Z-score values. For example, False Bay Park, Z_MK are $Z_{SPI-6} = -10.89$ $Z_{SPI-9} = -12.89$, $Z_{SPI-12} = -13.82$ and Z_MMK are $Z_{SPI-6} = -6.27$, $Z_{SPI-9} = -6.28$, $Z_{SPI-12} = -6.29$. The p-values of MK and MMK show the significance of the trends, with values way below 0.05 confirming statistically significant trends. In all cases except Riverview, the p-values are extremely low ($<< 0.05$), indicating strong evidence of significant decreasing trends in precipitation for all SPI periods. Both the MK and MMK tests confirm decreasing trends across all time scales, with the Z_MK and Z_MMK values becoming more negative as the SPI period increases, reflecting an intensifying downward trend over longer periods (from SPI-6 to SPI-12). For Riverview station, the results indicate an increasing trend with positive Z-score values, i.e. Z_MK are $Z_{SPI-6} = 2.85$, $Z_{SPI-9} = 3.84$, $Z_{SPI-12} = 4.59$ and Z_MMK are $Z_{SPI-6} = 1.19$, $Z_{SPI-9} = 2.16$, $Z_{SPI-12} = 2.29$. In general, all these results are consistent with those shown using the ITA (see Table 3). The Riverview station experience increasing trend because it is located closer to the coast, hence it is influenced by a combination of geographic, oceanic and climatic factors. For an example, this station could be influenced by the Agulhas Current, which flows southwards along the east coast of South Africa, bringing warm, moist air from the Indian Ocean, and thus enhancing evaporation that brings constant availability of moisture in the atmosphere.

**Table 2: Statistical summary of trend analysis for SPI-6, SPI-9, and SPI-12 using Mann-Kendall (MK) and Modified Mann-Kendall (MMK) tests.**

| False Bay Park | | | |
|---|---|---|---|
| **Variables** | **SPI-6** | **SPI-9** | **SPI-12** |
| $Z_{MK}$ | -10.89 | -12.89 | -13.82 |
| $p - value_{Mk}$ | < 0.00 | < 0.00 | < 0.00 |
| Decision ($Trend_{MK}$) | Decreasing | Decreasing | Decreasing |
| $Z_{MMK}$ | -6.27 | -6.28 | -6.29 |
| $p - value_{MMk}$ | $3.66 \times 10^{-10}$ | $3.35 \times 10^{-10}$ | $3.13 \times 10^{-10}$ |
| Decision ($Trend_{MK}$) | Decreasing | Decreasing | Decreasing |
| **Hlabisa Mbazwana** | | | |
| $Z_{MK}$ | -2.89 | -3.88 | -5.31 |
| $p - value_{Mk}$ | $3.77 \times 10^{-3}$ | $3.05 \times 10^{-4}$ | $1.10 \times 10^{-7}$ |
| Decision ($Trend_{MK}$) | Decreasing | Decreasing | Decreasing |
| $Z_{MMK}$ | -2.26 | -2.12 | -2.20 |
| $p - value_{MMk}$ | $2.39 \times 10^{-2}$ | $3.36 \times 10^{-2}$ | $2.78 \times 10^{-2}$ |



| Decision ($Trend_{MK}$) | Decreasing | Decreasing | Decreasing |
|---|---|---|---|
| **Pongolapoort Dam** | | | |
| $Z_{MK}$ | -7.19 | -8.74 | -9.83 |
| $p - value_{Mk}$ | $6.12 \times 10^{-13}$ | $< 0.00$ | $< 0.00$ |
| Decision ($Trend_{MK}$) | Decreasing | Decreasing | Decreasing |
| $Z_{MMK}$ | -8.22 | -5.44 | -6.51 |
| $p - value_{MMk}$ | $2.22 \times 10^{-16}$ | $5.40 \times 10^{-8}$ | $7.41 \times 10^{-11}$ |
| Decision ($Trend_{MK}$) | Decreasing | Decreasing | Decreasing |
| **Mkuze Game Reserve** | | | |
| $Z_{MK}$ | -3.66 | -5.54 | -6.67 |
| $p - value_{Mk}$ | $2.48 \times 10^{-4}$ | $2.99 \times 10^{-8}$ | $2.55 \times 10^{-11}$ |
| Decision ($Trend_{MK}$) | Decreasing | Decreasing | Decreasing |
| $Z_{MMK}$ | -2.44 | -2.79 | -2.22 |
| $p - value_{MMk}$ | $1.46 \times 10^{-2}$ | $5.13 \times 10^{-3}$ | $2.64 \times 10^{-2}$ |
| Decision ($Trend_{MK}$) | Decreasing | Decreasing | Decreasing |
| **Ingwavuma Manguzi** | | | |
| $Z_{MK}$ | -2.38 | -3.72 | -4.92 |
| $p - value_{Mk}$ | $1.72 \times 10^{-2}$ | $1.98 \times 10^{-4}$ | $8.72 \times 10^{-7}$ |
| Decision ($Trend_{MK}$) | Decreasing | Decreasing | Decreasing |
| $Z_{MMK}$ | -1.61 | -2.48 | -2.27 |
| $p - value_{MMk}$ | $1.08 \times 10^{-1}$ | $1.31 \times 10^{-2}$ | $2.29 \times 10^{-2}$ |
| Decision ($Trend_{MK}$) | Decreasing | Decreasing | Decreasing |
| **Riverview** | | | |
| $Z_{MK}$ | 2.85 | 3.84 | 4.59 |
| $p - value_{Mk}$ | $4.34 \times 10^{-3}$ | $1.25 \times 10^{-4}$ | $4.25 \times 10^{-6}$ |
| Decision ($Trend_{MK}$) | Increasing | Increasing | Increasing |
| $Z_{MMK}$ | 1.94 | 2.16 | 2.29 |
| $p - value_{MMk}$ | $5.12 \times 10^{-2}$ | $3.07 \times 10^{-2}$ | $2.19 \times 10^{-2}$ |
| Decision ($Trend_{MK}$) | Increasing | Increasing | Increasing |

**Table 3: The results of the trend analysis for SPI-6, SPI-9, and SPI-12 obtained through a two-tailed test at a significance level of 5% using ITA technique.**

| **False Bay Park** | | | |
|---|---|---|---|





| Variables | SPI-6 | SPI-9 | SPI-12 |
|---|---|---|---|
| Slope | $-3.51 \times 10^{-3}$ | $-1.14 \times 10^{-3}$ | $-4.49 \times 10^{-3}$ |
| Indicator | -20.08 | -20.12 | -20.07 |
| $\pm$CI at 95% | $\pm9.24 \times 10^{-5}$ | $\pm7.52 \times 10^{-5}$ | $\pm6.82 \times 10^{-5}$ |
| **Hlabisa Mbazwana** | | | |
| Slope | $-1.68 \times 10^{-3}$ | $-2.31 \times 10^{-3}$ | $-1.86 \times 10^{-3}$ |
| Indicator | $-20.52$ | $-20.72$ | $-20.64$ |
| $\pm$CI at 95% | $\pm6.81 \times 10^{-5}$ | $\pm9.35 \times 10^{-5}$ | $\pm7.15 \times 10^{-5}$ |
| **Pongolapoort Dam** | | | |
| Slope | $2.26 \times 10^{-3}$ | $-2.88 \times 10^{-3}$ | $-3.34 \times 10^{-3}$ |
| Indicator | $-19.27$ | $-19.40$ | $-19.55$ |
| $\pm$CI at 95% | $\pm2.22 \times 10^{-5}$ | $\pm3.62 \times 10^{-5}$ | $\pm6.72 \times 10^{-5}$ |
| **Mkuze Game Reserve** | | | |
| Slope | $-2.00 \times 10^{-3}$ | $-3.04 \times 10^{-3}$ | $-3.80 \times 10^{-3}$ |
| Indicator | $-20.09$ | $-20.22$ | $-20.25$ |
| $\pm$CI at 95% | $\pm2.81 \times 10^{-3}$ | $\pm4.67 \times 10^{-3}$ | $\pm4.40 \times 10^{-3}$ |
| **Ingwavuma Manguzi** | | | |
| Slope | $-1.61 \times 10^{-3}$ | $-2.26 \times 10^{-3}$ | $-2.88 \times 10^{-3}$ |
| Indicator | $-21.96$ | $-21.05$ | $-20.77$ |
| $\pm$CI at 95% | $\pm6.81 \times 10^{-5}$ | $1.01 \pm\times 10^{-5}$ | $\pm1.19 \times 10^{-5}$ |
| **Riverview** | | | |
| Slope | $1.69 \times 10^{-3}$ | $2.19 \times 10^{-3}$ | $2.37 \times 10^{-3}$ |
| Indicator | 22.54 | 22.22 | 21.86 |
| $\pm$CI at 95% | $\pm1.54 \times 10^{-5}$ | $\pm1.35 \times 10^{-5}$ | $\pm1.56 \times 10^{-5}$ |

### 3.3. SPI Time Series Forecasting Results

The study proposes a hybrid model that applies the Savitzky-Golay (SG) filter to raw SPI data to reduce noise and improve forecasting analysis. To demonstrate the effectiveness of the SG filter, appropriate parameters such as window size and polynomial order were selected through trial and error using data from the study sites (Sibiya et al., 2024). A window size of 21 and a polynomial order of 5 were chosen for smoothing. Figure 9 shows how the SG filter effectively tracks the general trend while preserving the shape of peaks and minimizing noise. This filter was applied to different time scales of the SPI time series. It autonomously calibrates according to peak distribution, exhibiting optimal performance, particularly with asymmetric peaks, while preserving peak height integrity. The application of the SG filter effectively mitigates short-term fluctuations and eliminates noise from the time series resulting in cleaner data, thereby enhancing the reliability of the subsequent decomposition process. By reducing noise, decomposition techniques can more accurately capture the authentic underlying patterns and components within the data.



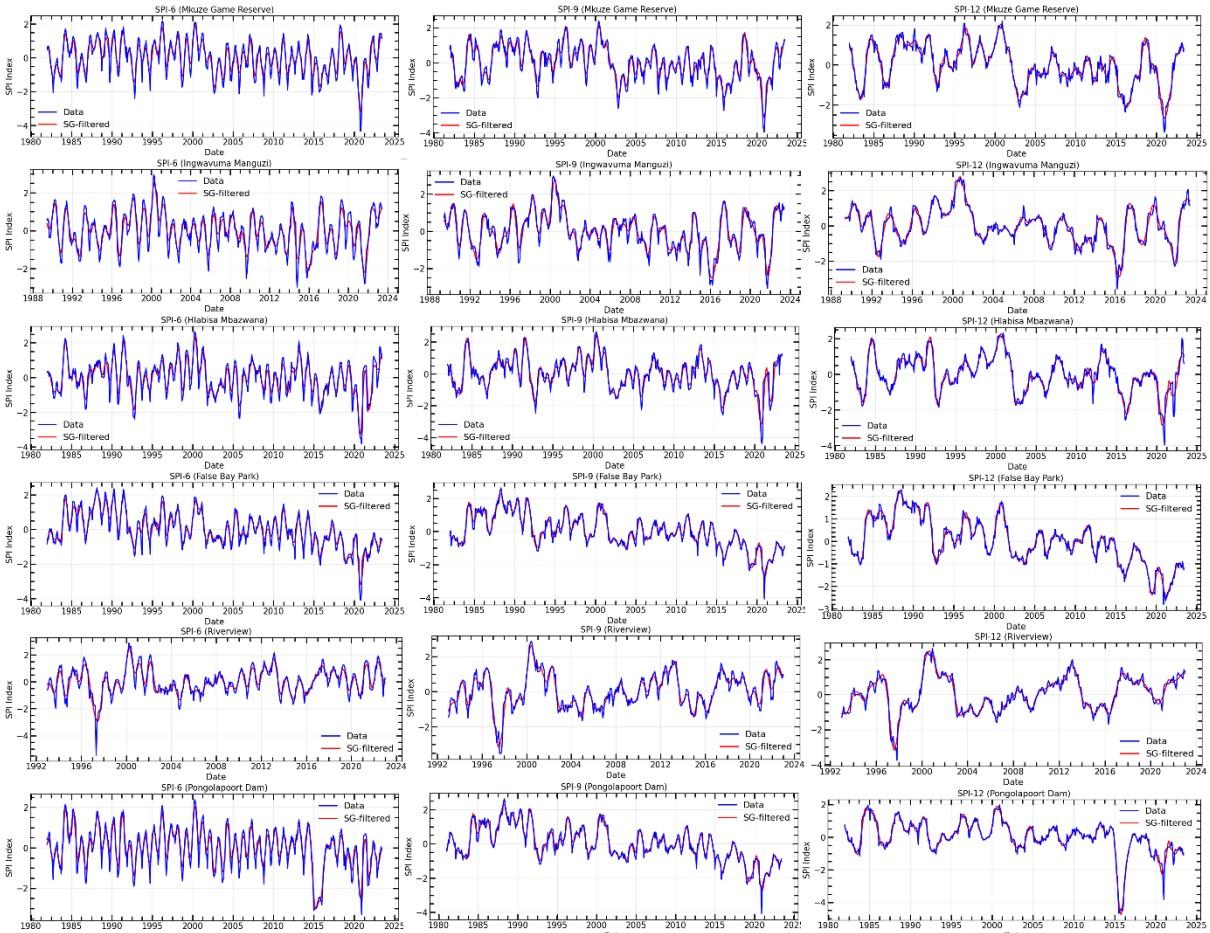

**Figure 9: SPI signals smoothed by Savitzky-Golay (SG). The blue line represents the original signal and red line is the SG smoothened signal.**

After applying a Savitzky-Golay filter to the series, the CEEMDAN algorithm decomposes the filtered SPI series into six subseries with different amplitudes and frequencies. The results from the False Bay Park station are utilized here as an illustration to prevent repetition. In these results, the decomposed set of time series consists of five IMF components and a residual component, as shown in Figure 10 (for all time scales). During the decomposition process, white Gaussian noise is added to create noisy signals. The original sequence exhibits high nonlinearity and nonstationarity, with the frequency of the IMF components gradually decreasing. Figure 10 depicts this gradual decrease in frequency as the order of the IMF components increases. As each IMF is further decomposed, it becomes less volatile and cyclical, which aligns with the characteristics of the decomposed IMF. Therefore, by predicting each IMF and the residual, the forecast precision can be enhanced. A forecasting model is then constructed for each component, and the prediction results are obtained by summing up the outputs of all predicted components.





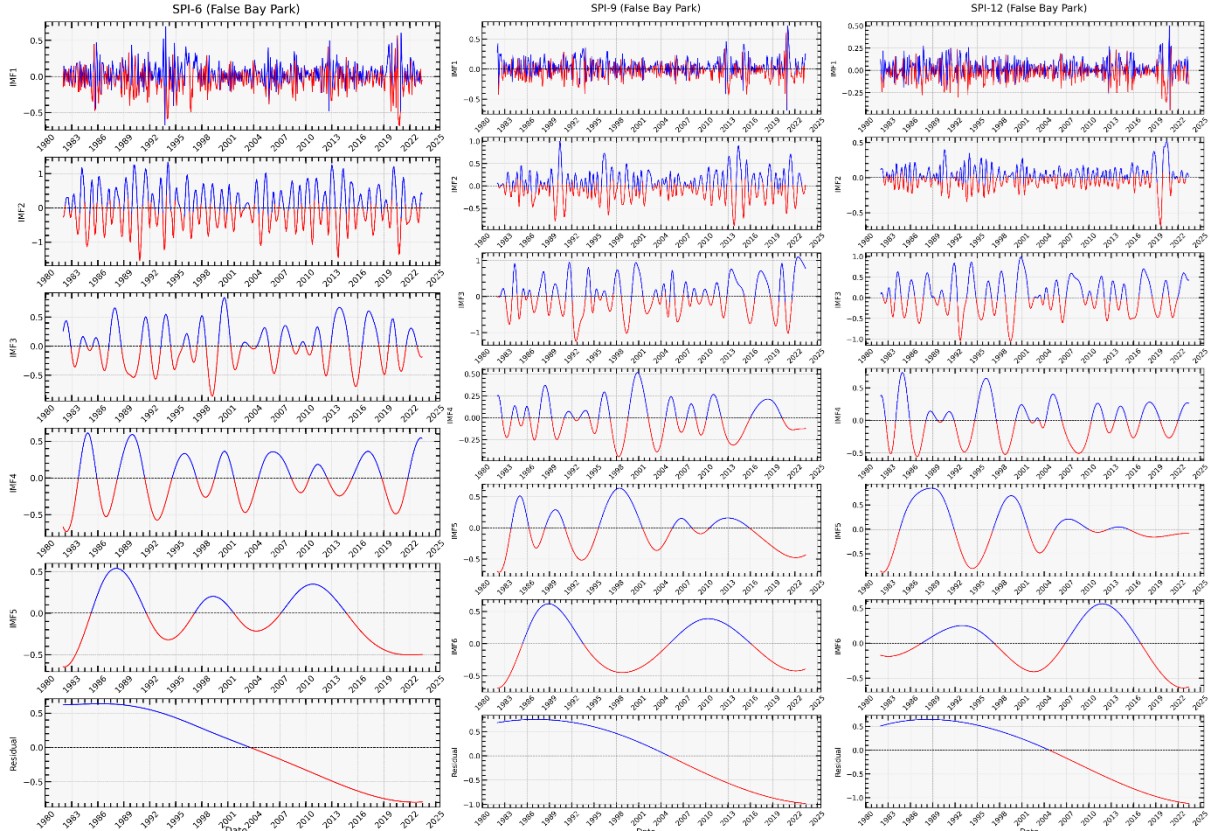


**Figure 10: Decomposition of Smoothed SPI-6, SPI-9 and SPI-12 Index Using CEEMDAN: Each IMF represents different frequency components of the SPI index, from high-frequency oscillations (IMF1) to low-frequency trends (IMF5), showing the variability in precipitation patterns over the years from 1980 to 2023.**

The models in Table 4 were compared for their prediction ability before and after time series decomposition in this research. The objective was to determine if smoothing and decomposing time series improves the model's prediction performance. Figures 11–16 show a comparison of the various models' prediction outcomes using the Taylor diagram. In general, all the models accurately replicate the original SPI time series at all timescales (refer to Figure 11 - 16) in terms of the time series plot. However, the SG-CEEMDAN-ARIMA-LSTM model (shown in red) appears to have the closest fit to the data, displaying

superior accuracy across different phases, particularly in extreme values. Nonetheless, the hybrid models (SG-ARIMA-LSTM, CEEMDAN-ARIMA-LSTM, and SG-CEEMDAN-ARIMA-LSTM) show better precision in capturing peaks, rapid transitions and troughs compared to the standalone LSTM or ARIMA models. Table 4 displays an assessment of the predictive performance metrics of several models utilising RMSE, $R^2$, and DS. As the period extends, the RMSE values decrease, however the DS and $R^2$ values typically enhance (see Table 4). This indicates that the models' predictive accuracy

progressively enhances with an extended duration, reaching its highest point at the 12-month interval. In terms of RMSE, the SG-CEEMDAN-ARIMA-LSTM model outperforms the others, exhibiting the lowest error values across all indices. For





example, Riverview station, 0.2165 for SPI-6, 0.0921 for SPI-9, and 0.0566 for SPI-12. This indicates that this model has the smallest prediction error, making it the most accurate in terms of error reduction. Concerning $R^2$, which measures how well the model explains the variance in the data, SG-CEEMDAN-ARIMA-LSTM again leads with the highest values: 0.9602 for

SPI-6, 0.9846 for SPI-9, and 0.9939 for SPI-12. This shows that the model provides the best fit to the data. The CEEMDAN-ARIMA-LSTM model is the second-best performer, also exhibiting impressive results, particularly in $R^2$, where it achieves higher values of 0.9483 for SPI-6, 0.9751 for SPI-9, and 0.9933 for SPI-12. The SG-ARIMA-LSTM model is the third-best hybrid performer, with RMSE values of 0.2262 for SPI-6, 0.1051 for SPI-9, and 0.05639 for SPI-12. The SG-ARIMA-LSTM model is the third-best performer, also exhibiting impressive results, particularly in $R^2$, where it achieves higher values of

0.9392 for SPI-6, 0.9763 for SPI-9, and 0.9904 for SPI-12. The SG-ARIMA-LSTM model is the third-best hybrid performer, with RMSE values of 0.2597 for SPI-6, 0.1157 for SPI-9, and 0.0567 for SPI-12. In general, these results highlight the efficacy of hybrid models, particularly those incorporating SG and CEEMDAN processes, in improving predictive accuracy across multiple timescales of SPI, particularly for the SG-CEEMDAN-ARIMA-LSTM model. These results are consistent with Taylor diagram (see Figure 11 - 16) which indicates a significant improvement in prediction accuracy after incorporating the

SG and CEEMDAN signal decomposition technique as the hybrid model exhibits superior performance in terms of prediction accuracy across all timescales, surpassing other models. This suggests that the inclusion of these techniques enhances the models' ability to capture both short-term and long-term dependencies, thus making them more robust for drought prediction purposes. Therefore, this hybrid model appears to be the most effective for drought prediction in this analysis. These findings highlight the superiority of the proposed hybrid model in enhancing drought prediction accuracy compared to standalone

approaches.

**Table 4. Performance measures for the comparison of observed and forecasted data of the models for SPI-6, SPI-9 and SPI-12 across various lead times using statistical criteria.**

| False Bay Park | | | | | | | | |
|---|---|---|---|---|---|---|---|---|
| Model | SPI-6 | | | SPI-9 | | | SPI-12 | | |
| | RMSE | $R^2$ | DS | RMSE | $R^2$ | DS | RMSE | $R^2$ | DS |
| ARIMA | 0.3504 | 0.8435 | 0.8426 | 0.2431 | 0.8976 | 0.8525 | 0.1689 | 0.9421 | 0.8426 |
| LSTM | 0.3128 | 0.9111 | 0.8327 | 0.2416 | 0.9521 | 0.8723 | 0.1626 | 0.9821 | 0.8519 |
| ARIMA-LSTM | 0.2476 | 0.9194 | 0.8327 | 0.1650 | 0.9531 | 0.8723 | 0.0507 | 0.9952 | 0.9009 |
| SG-ARIMA-LSTM | 0.2056 | 0.9458 | 0.8030 | 0.1348 | 0.9687 | 0.8218 | 0.0571 | 0.9940 | 0.9009 |
| C-A-L | 0.2182 | 0.9375 | 0.8713 | 0.0978 | 0.9834 | 0.8218 | 0.0496 | 0.9953 | 0.8911 |
| SG-C-A-L | 0.1835 | 0.9650 | 0.8416 | 0.1631 | 0.9836 | 0.8317 | 0.0349 | 0.9957 | 0.8941 |
| Mkuze Game Reserve | | | | | | | | |



| | | | | | | | | | |
|---|---|---|---|---|---|---|---|---|---|
| ARIMA | 0.3752 | 0.8642 | 0.8419 | 0.3475 | 0.8957 | 0.8792 | 0.2202 | 0.9697 | 0.8730 |
| LSTM | 0.3474 | 0.9121 | 0.8822 | 0.3354 | 0.9178 | 0.8030 | 0.1523 | 0.9890 | 0.8733 |
| ARIMA-LSTM | 0.3160 | 0.9273 | 0.8416 | 0.1561 | 0.9823 | 0.8218 | 0.1079 | 0.9926 | 0.8730 |
| SG- ARIMA-LSTM | 0.2307 | 0.9624 | 0.8515 | 0.1548 | 0.9825 | 0.8317 | 0.08252 | 0.9951 | 0.8019 |
| C-A-L | 0.1969 | 0.9726 | 0.8317 | 0.1430 | 0.9850 | 0.8515 | 0.04497 | 0.9986 | 0.9208 |
| SG-C-A-L | 0.1818 | 0.9742 | 0.8515 | 0.1232 | 0.9892 | 0.8617 | 0.04217 | 0.9990 | 0.9208 |
| Pongolapoort Dam | | | | | | | | | |
| ARIMA | 0.4470 | 0.8797 | 0.8624 | 0.2993 | 0.9668 | 0.8119 | 0.1918 | 0.9763 | 0.8733 |
| LSTM | 0.4470 | 0.8962 | 0.8732 | 0.2873 | 0.9467 | 0.8238 | 0.1824 | 0.9851 | 0.8829 |
| ARIMA-LSTM | 0.4121 | 0.8969 | 0.8822 | 0.2599 | 0.9588 | 0.8921 | 0.1638 | 0.9862 | 0.8432 |
| SG- ARIMA-LSTM | 0.2224 | 0.9617 | 0.8019 | 0.2064 | 0.9803 | 0.8515 | 0.0686 | 0.9969 | 0.8119 |
| C-A-L | 0.2132 | 0.9649 | 0.8822 | 0.1572 | 0.9850 | 0.8218 | 0.0639 | 0.9975 | 0.8019 |
| SG-C-A-L | 0.1453 | 0.9839 | 0.8824 | 0.1429 | 0.9858 | 0.8911 | 0.0635 | 0.9978 | 0.8921 |
| Hlabisa Mbazwana | | | | | | | | | |
| ARIMA | 0.4704 | 0.8347 | 0.8624 | 0.4234 | 0.8698 | 0.8921 | 0.2321 | 0.9556 | 0.8142 |
| LSTM | 0.3617 | 0.9041 | 0.8327 | 0.2163 | 0.9672 | 0.8119 | 0.1566 | 0.9806 | 0.8317 |
| ARIMA-LSTM | 0.3269 | 0.9369 | 0.8515 | 0.2139 | 0.9677 | 0.8218 | 0.1457 | 0.9813 | 0.8426 |
| SG- ARIMA-LSTM | 0.3011 | 0.9355 | 0.8416 | 0.1829 | 0.9747 | 0.8317 | 0.08540 | 0.9935 | 0.8218 |
| C-A-L | 0.2497 | 0.9592 | 0.8218 | 0.1662 | 0.9792 | 0.8218 | 0.0825 | 0.9949 | 0.9009 |
| SG-C-A-L | 0.1921 | 0.9795 | 0.8614 | 0.1332 | 0.9866 | 0.8218 | 0.07416 | 0.9952 | 0.9029 |
| Ingwavuma Manguzi | | | | | | | | | |
| ARIMA | 0.4123 | 0.8716 | 0.8571 | 0.2706 | 0.9442 | 0.8750 | 0.2052 | 0.9784 | 0.8619 |
| LSTM | 0.3843 | 0.8931 | 0.8738 | 0.2524 | 0.2524 | 0.8691 | 0.1614 | 0.9828 | 0.8095 |
| ARIMA-LSTM | 0.3458 | 0.9044 | 0.8095 | 0.2428 | 0.9695 | 0.8541 | 0.8541 | 0.9847 | 0.8215 |
| SG- ARIMA-LSTM | 0.2767 | 0.9397 | 0.8076 | 0.2001 | 0.9724 | 0.8809 | 0.0815 | 0.9958 | 0.8929 |
| C-A-L | 0.2536 | 0.9503 | 0.8095 | 0.1945 | 0.9719 | 0.8214 | 0.0739 | 0.9972 | 0.9167 |
| SG-C-A-L | 0.2314 | 0.9565 | 0.8214 | 0.1575 | 0.9823 | 0.8809 | 0.0634 | 0.9978 | 0.8809 |
| Riverview | | | | | | | | | |
| ARIMA | 0.4375 | 0.8132 | 0.8106 | 0.1708 | 0.9474 | 0.8038 | 0.1137 | 0.9570 | 0.7973 |





| | | | | | | | | |
|---|---|---|---|---|---|---|---|---|
| LSTM | 0.3212 | 0.8510 | 0.8108 | 0.1537 | 0.9400 | 0.8108 | 0.0982 | 0.9705 | 0.8273 |
| ARIMA-LSTM | 0.2874 | 0.8767 | 0.8378 | 0.1314 | 0.9706 | 0.9595 | 0.0558 | 0.9934 | 0.9189 |
| SG-ARIMA-LSTM | 0.2262 | 0.9392 | 0.8243 | 0.1051 | 0.9763 | 0.8243 | 0.05639 | 0.9904 | 0.8108 |
| C-A-L | 0.2597 | 0.9483 | 0.8738 | 0.1157 | 0.9751 | 0.9324 | 0.05674 | 0.9933 | 0.9459 |
| SG-C-A-L | 0.2165 | 0.9602 | 0.8919 | 0.09214 | 0.9846 | 0.9324 | 0.05664 | 0.9939 | 0.9189 |

Note: C-A-L = CEEMDAN-ARIMA-LSTM


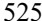

**Figure 11: The time series of observations and hybrid forecasting models for SPI prediction (Left) and their Taylor diagram plots at different timescales (Right) for SPI-6, SPI-9, and SPI-12 of Riverview meteorological station.**






**Figure 12: The time series of observations and hybrid forecasting models for SPI prediction (Left) and their Taylor diagram plots at different timescales (Right) for SPI-6, SPI-9, and SPI-12 of Hlabisa Mbazwana meteorological station.**





**Figure 13: The time series of observations and hybrid forecasting models for SPI prediction (Left) and their Taylor diagram plots**
**at different timescales (Right) for SPI-6, SPI-9, and SPI-12 of Ingwavuma Manguzi meteorological station.**





**Figure 14: The time series of observations and hybrid forecasting models for SPI prediction (Left) and their Taylor diagram plots at different timescales (Right) for SPI-6, SPI-9, and SPI-12 of Mkuze Game Reserve meteorological station.**







**Figure 15: The time series of observations and hybrid forecasting models for SPI prediction (Left) and their Taylor diagram plots at different timescales (Right) for SPI-6, SPI-9, and SPI-12 of Pongolapoort dam meteorological station**







**Figure 16: The time series of observations and hybrid forecasting models for SPI prediction (Left) and their Taylor diagram plots at different timescales (Right) for SPI-6, SPI-9, and SPI-12 of False Bay Park meteorological station.**


## 4.   Discussion

In this study, we utilized the Mann-Kendall and Modified Mann-Kendall tests to determine the drought trends index in meteorological variables within the basin. The MK and MMK trend methods showed a significant decrease in all SPI time scales based on rainfall data from five stations; however, the district, except for the Riverview station, showed an increasing

trend in the uMkhanyakude district. The study's findings align with prior research by Kganvago et al. (2021) and Ngwenya et al. (2024). Ngwenya et al. (2024) performed a study employing the Mann-Kendall test to evaluate the SPI values at a 5% significance level, revealing sustained drought conditions in the Western Cape area. Kganvago et al. (2021) indicated a notable decline in drought conditions in the Western Cape area of South Africa. We have also employed the ITA, which enhances the MK and MMK tests in identifying trends, and the results underscore the importance of comprehending drought conditions.





The findings of our analysis validate previous research by Naik and Abiodun (2020), highlighting the necessity of performing trend studies on drought indicators to investigate the impacts of climate change. The study underscores the essential function of SPI as a primary variable in monitoring and forecasting droughts in the region, and its potential to mitigate the detrimental impacts of droughts and water scarcity in the uMkhanyakude district in the future. The objective was to determine if the model's predictive performance is enhanced by smoothing and deconstructing time series data.


According to the statistical metrics in Table 4 and Taylor diagram (see Figure 11 - 16), highlight the effectiveness of hybrid models that incorporate filter and signal decomposition techniques (SG and CEEMDAN) in improving prediction accuracy, particularly for drought forecasting. These findings support other research (Taylan et al., 2021; Elbeltagi et al., 2023; Rezaiy and Shabri 2024b) highlighting the superior accuracy of hybrid drought forecasting models relative to an individual models.

For example, Taylan et al. (2021) developed a hybrid model to forecast drought using precipitation data from Çanakkale, Gökçeada, and Bozcaada stations between 1975 and 2010. The study found that the hybrid models, which incorporated preprocessing techniques, performed better. Elbeltagi et al. (2023) utilized a hybrid model to estimate the SPI for 3, 6, and 12-month drought periods from 2000 to 2019. The findings demonstrated that RSS-M5P model yielded the most precise SPI predictions, with $MAE = 0.497$, $RMSE = 0.682$, $RAE = 81.88$, $RRSE = 87.22$, and $R^2 = 0.507$ for SPI-3; $MAE = 0.452$,

$RMSE = 0.717$, $RAE = 69.76$, $RRSE = 85.24$, and $R^2 = 0.402$ for SPI-6 and $MAE = 0.294$, $RMSE = 0.377$, $RAE = 55.79$, $RRSE = 59.57$, and $R^2 = 0.783$ for SPI-12. The models employed to analyse drought in Jaisalmer, Rajasthan, yielded the most effective results, exceeding those of RSS-RF and RSS-RT. Additionally, Rezaiy and Shabri (2024b) introduced a W-EEMD-ARIMA model for drought prediction. This model utilises monthly precipitation data from Kabul spanning 1970 to 2019. The $R^2$ value was 0.9946, the MAPE was 18.9674, the RMSE was 0.0736, the MAE was 0.0575, and the SPI-12 validation

indicated that our model was accurate. The outcomes obtained here surpassed those of the ARIMA, Wavelet-ARIMA, and EEMD-ARIMA models in terms of raw data (RMSE: 0.0858, MAE: 0.0660, MAPE: 24.5411, $R^2$ : 0.9925), analytical method (MAE: 0.1874, MAPE: 60.0220, $R^2$ : 0.9361), and maximum likelihood estimation (RMSE: 0.1002, MAE: 0.0691, MAPE: 23.7122, $R^2$ : 0.9898). During the SPI-3, SPI-6, and SPI-9 periods, our hybrid model consistently outperformed other models. Our proposed hybrid model surpasses ARIMA, Wavelet-ARIMA, and EEMD-ARIMA in enhancing the precision of drought

predictions, as evidenced by this data.

In terms of term forecasting accuracy, the hybrid models, SG-CEEMDAN-ARIMA-LSTM in particular consistently surpassed all other models across all SPI timescales, according to a comparison of this study's results with previous research. All models successfully reproduced the original SPI time series. With the range values of RMSE of 0.1453 - 0.2314 for SPI-6, 0.0921 – 0.1631 for SPI-9, and 0.0349 – 0.07416 for SPI-12, and the highest $R^2$ values of 0.9565 - 0.9839 for SPI-6, 0.9836 - 0.9892

for SPI-9, and 0.9939 - 0.9990 for SPI-12 across all timescales, the SG-CEEMDAN-ARIMA-LSTM model showed the most proficiency in capturing extreme values and rapid transitions. That these methods, when combined, improve the models' capacity to represent drought in uMkhanyakude district, both in the short and long term, is supported by the data. This makes




the models far better at foretelling when droughts will occur. In light of the foregoing, our study provides useful information regarding the use of the hybrid SG-CEEMDAN-ARIMA-LSTM model to the forecasting of meteorological droughts.


## 5. Conclusion

This study examined the trends in the Standardised Precipitation Index (SPI) over different timescales (SPI-6, SPI-9, and SPI-12) utilising the Mann-Kendall (MK), modified Mann-Kendall (MMK) test, and the innovative trend analysis (ITA) protocol. The monthly rainfall data from uMkhanyakude district, South Africa, covering the years 1980 to 2023, was used for these

calculations. Rainfall has been trending downward at a 95% confidence level, according to the MK and MMK tests. The ITA results supported these findings as well, revealing a declining trend with the most of data points going below the 1:1 line. In order to predict SPI data over various timescales, this research also used LSTM and autoregressive integrated moving average (ARIMA) models. Researchers used a hybrid model that combines the SG-CEEMDAN processing method with the ARIMA-LSTM model to enhance the precision of SPI forecasts. They also used SG filtering and full ensemble empirical mode

decomposition with adaptive noise (CEEMDAN). Figures 11–16 and Table 4 display results of a thorough comparison examination of the forecast outcomes. The results revealed that the inclusion of preprocessing techniques (SG filtering, CEEMDAN, and SG-CEEMDAN) significantly improved the model performance in forecasting SPI at all timescales. The performance consistently increased with higher timescales, potentially due to lower noise levels. Across different timescales, SG and CEEMDAN combined hybrid model consistently outperformed the individual models. Notably, the CEEMDAN-

ARIMA-LSTM model outperformed the SG-ARIMA-LSTM model at all timescales, while the SG-CEEMDAN-ARIMA-LSTM model consistently exhibited the lowest root mean square error (RMSE) values across all indices. These results demonstrate that combining SG-CEEMDAN with ARIMA-LSTM has the potential to significantly enhance the accuracy of meteorological drought forecasting.

The principal conclusion of the study is that ARIMA-LSTM, in conjunction with SG, CEEMDAN, and SG-CEEMDAN,

serves as an effective instrument for early warning systems and meteorological drought prediction. The proposed methodology in this paper serves as a framework for modeling complex meteorological phenomena such as drought, particularly pertinent in semi-arid regions. Enhancing model performance and creating efficient models for weather forecasting can be achieved through techniques that address data noise, nonlinearity, and nonstationarity. To enhance water resource management, make informed decisions regarding agricultural output and tourism management, and establish regulations, it is essential to acquire

extremely effective models for drought prediction. The omission of exogenous environmental variables in the SG-CEEMDAN-ARIMA-LSTM model represents a significant drawback of the study. The model's forecast accuracy and real-world application are limited by disregarding these exogenous effects, which can substantially affect drought conditions. Future study should aim to include external variables, including temperature, soil moisture, vegetation indices, and anthropogenic factors such as land use and water management, to improve the model's efficacy. This integration would provide a more thorough



comprehension of drought dynamics, hence improving the model's accuracy and dependability in drought predictions. Additionally, it is essential to investigate alternate decomposition methods, such as enhanced CEEMDAN (iCEEMDAN), which may provide significant insights.

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
