# Peer review of "Meteorological Drought Trend Analysis and Forecasting Using a Hybrid SG-CEEMDAN-ARIMA-LSTM Model Based on SPI from Rain Gauge Data"

_EGUsphere, 2025_

## Referee Comment (RC2)

This manuscript presents a promising and methodologically innovative approach to drought modeling, particularly with its integration SG-CEEMDAN-ARIMA-LSTM. However, to meet the rigor expected by journal, the following revisions would strengthen its statistical grounding and reproducibility:

1- It is recommended incorporating the most recent literature (particularly studies published in 2024-2025) on hybrid drought forecasting methods to ensure the methodology reflects current advances in the field

2- The upper bound of the cumulative probability function $H(x)$ in Equation 21 is incorrectly defined to include the value 1. This leads to an undefined expression:

$$\ln\left(\left(\frac{1}{1-H(x)}\right)^2\right)$$

when $H(x)=1$, which evaluates to $\ln(\infty)$. This is mathematically invalid and computationally dangerous, as it can cause overflow or undefined behavior in implementation. The formula must be constrained to $H(x)\in(0,1)$, not $[0,1]$. This critical issue must be addressed and corrected in the manuscript before further consideration.

3- Even though MMK (Modified Mann-Kendall) is applied, the paper doesn't explain how the lag selection was determined for autocorrelation adjustments. Additionally, there are no ACF or PACF plots provided to support the chosen ARIMA model order.

4-Before using ARIMA, the study should have checked for stationarity using tests like ADF, but this step isn't mentioned. Also, it's unclear whether the data was normalized or scaled before being fed into the LSTM model or not.

5- While the baseline models (ARIMA, LSTM, and CEEMDAN-LSTM) provide useful predictions, the study lacks a formal statistical comparison such as ANOVA to objectively assess their accuracy differences.

---

## Author Comment (AC1)

**Response to Reviewer 1 Comments**

General reply to the reviewer*: We would like to express our gratitude to the reviewer for their invaluable suggestions and comments. These comments have indicated to be very important for strengthening our manuscript draft. It gave us an opportunity to look at our work in a different way.*

The manuscript presents an interesting and relevant study on forecasting SPI using a hybrid model that combines existing methodologies in a novel way. The approach, which integrates signal decomposition (SG, CEEMDAN) with traditional (ARIMA) and deep learning (LSTM) techniques, addresses a crucial topic with significant potential impact, particularly for data-scarce regions like uMkhanyakude, South Africa. Although the study does not introduce entirely new methods, the unique combination and application of established techniques offer valuable insights and could help advance drought forecasting.

Major Comments:

- Terminological Precision and Focus on Introduction: The manuscript frequently uses the term "drought" without clearly specifying its type until late in the introduction. The initial sections should explicitly state that the study focuses on meteorological drought, as defined by the Standardized Precipitation Index (SPI), which only reflects precipitation. This clarification is crucial to avoid confusion and help set the stage for the study's objectives, contributions, and context within the broader field of drought research. Additionally, the introduction should discuss the scope and limitations, particularly noting that the study is a new method for SPI forecasting.

**Reply:** *We would like to thank the reviewer for this comment. This is now done. We have revised the introduction and address the reviewers' comments.*

- Methodological Framing and Clarity: The Methods section should provide a clearer and more focused explanation of the hybrid modeling framework. This includes: the rationale behind combining SG filtering with CEEMDAN decomposition prior to modeling; how the decomposition into intrinsic mode functions (IMFs) enhances forecasting accuracy, as indicated by the improved RMSE and values shown in Table 4; the stepwise integration of the ARIMA and LSTM models on decomposed components, and how these components are recombined for final predictions; the comparative advantages of this hybrid method over standalone models or simpler combinations, evidenced by the superior performance of the SG-CEEMDAN-ARIMA-LSTM model across all SPI timescales, as shown in Figures 11-16. While a diagram is present, these aspects should be emphasized to highlight the novel integration strategy, rather than detailing standard approaches like ARIMA or LSTM. These well-known methods can be briefly summarized, with detailed descriptions moved to the appendix to lighten the paper and assist the reader.

**Reply**: *We would like to thank the reviewer for this comment. This was done. We have revised the methodology and address the reviewers' comments.*

- Streamlining Content: To improve readability, consider moving detailed descriptions of well-known methods to an appendix. This will allow the main text to focus more on the innovative aspects of the study and its implications.

**Reply:** *We would like to thank the reviewer for this important comment. We considered explaining details of each methodology that assisted us built our new proposed SG-CEEMDAN-ARIMA-LSTM hybrid model because details of all these methodologies seem to be very important for the readers. To create a flow and a better readability of our materials and method section, we have added a section that introduce what we are doing in the section and why. We did this to show that such a structured presentation ensures transparency in model development and establishes a comprehensive methodological framework for the proposed forecasting system (SG-CEEMDAN-ARIMA-LSTM).*

- Justification for Methodological Choices: While the manuscript acknowledges the limitations of SPI, it should provide a more robust justification for its selection over SPEI, particularly under climate change conditions. Addressing this could strengthen the methodological rationale by discussing factors such as data availability or regional relevance.

**Reply***: We would like to thank the reviewer for this comment. We have revised and address the reviewers' comments. Some of the important points about using SPI is that:*

- *Data simplicity: Only precipitation data needed; SPEI requires reliable temperature data.*
- *Less uncertainty: PET estimates can be inaccurate, especially in regions with limited meteorological stations.*
- *Consistency in long-term studies: Historical precipitation data may go back decades or more, while temperature and PET records may be shorter or less reliable.*
- *Focus on rainfall-driven drought: In regions where evapotranspiration is not the dominant driver, SPI is sufficient.*
- *Comparability: SPI is widely reported globally; easier to benchmark across regions and studies.*

- Literature Review Organization: The literature review should be reorganized to group studies thematically, highlighting insights that motivate the proposed model and clarifying the research gap that this study aims to address. This will provide a clearer context for the study's contributions.

*We would like to thank the reviewer for this comment. We have revised the literature review and address the reviewers' comments.*

- Abstract and Title Refinement: The abstract should be concise and precise, clearly outlining the study's objectives and methods. Similarly, consider revising the title to avoid redundancy and focus on the paper's core contributions.

*Reply: We would like to thank the reviewer for this comment. We have revised the abstract, tittle and address the reviewers' comments.*

In summary, the manuscript is potentially interesting and relevant, offering valuable insights through its novel combination of established methodologies. However, it would benefit from a rewrite to clarify key sections in the Introduction and Methods, and from streamlining redundant content to enhance readability and focus.

*Reply: We would like to express our gratitude to the reviewer for their invaluable suggestions and comments.*

---

## Author Comment (AC2)

**Response to Reviewer 3 Comments**

General reply to the reviewer*: We would like to express our gratitude to the reviewer for their invaluable suggestions and comments. These comments have indicated to be very important for strengthening our manuscript draft. It gave us an opportunity to look at our work in a different way.*

The manuscript addresses the topic related to drought trend analysis and forecasting, and it appears that the authors have invested considerable effort in applying a combination of statistical and machine learning techniques. However, the manuscript suffers from several critical issues that need to be addressed before it can be considered for publication. The methodology section lacks clarity, the figures are not adequately formatted for readability, and the introduction is poorly structured. I recommend major revisions to improve the overall clarity. Here are some potentially helpful suggestions:

**Introduction**:

The first paragraph could benefit from improved focus and clearer logic. While it introduces the general impacts of drought, the core message is somewhat diluted. The second paragraph seems only loosely connected to the main theme of the study. I suggest the authors focus more specifically on summarizing the strengths and limitations of various drought prediction methods, rather than listing a large number of references without clear synthesis. Additionally, the third and fourth paragraphs appear closely related and might be more effective if combined.

*We would like to thank the reviewer for this comment. Done, we have revised the introduction and addressed the reviewers' comments.*

**Method**:

The authors spend a significant amount of time explaining the algorithms or working principles of SG, CEEMDAN, ARIMA, and LSTM models, which are well-known techniques. What I would like to see is how these models are integrated together—whether they form a framework or are coupled in some way. I would also like to know how the parameters for these models were set.

**Reply:** *We would like to thank the reviewer for this comment. This was included and explained under the methodology section.*

**Results**

1. The figure. 6 is not properly aligned and appear to be more suited for a report format. And I consider this figure is not the results of this paper.

**Reply:** *We acknowledge the reviewer's concern that Figure 6 seems better suited for a report format and does not directly present results. However, this figure is essential for the study as it provides a preliminary visualization of daily and monthly rainfall across all meteorological stations. Inspecting the raw and aggregated rainfall data is critical before computing the SPI and conducting trend analysis to understand temporal patterns, seasonal cycles, and extreme*

*variability. The figure ensures transparency by showing the underlying dataset used for subsequent analysis and highlights the need for data quality checks before applying SPI calculation and forecasting models.*

2. Lines 490 – Lines 520: It appears that in-situ data from 1980–2014 was used for training, and 2015–2023 for testing. This setup raises concerns about potential overfitting. To further demonstrate the model's generalizability, I suggest the authors consider adding a transfer prediction experiment.

*Reply: We would like to thank the reviewer for this comment. We did consider overfitting when using Bayesian optimization in the models by using cross-validation instead of a single train–test split during optimization and include an early stopping rule for iterative models. This model was fitted on 6 different stations on the data with the same length assuming that was enough for the model's generalizability.*

3. Given that parameter selection can significantly affect model performance, a more detailed explanation of the tuning procedures for each model would strengthen the methodological transparency.

*Reply: We would like to thank the reviewer for this comment. This was included and explained under the results section.*

---

## Author Comment (AC3)

**Response to Reviewer 2 Comments**

General reply to the reviewer*: We would like to express our gratitude to the reviewer for their invaluable suggestions and comments. These comments have indicated to be very important for strengthening our manuscript draft. It gave us an opportunity to look at our work in a different way.*

This manuscript presents a promising and methodologically innovative approach to drought modeling, particularly with its integration SG-CEEMDAN-ARIMA-LSTM. However, to meet the rigor expected by journal, the following revisions would strengthen its statistical grounding and reproducibility:

1- It is recommended incorporating the most recent literature (particularly studies published in 2024-2025) on hybrid drought forecasting methods to ensure the methodology reflects current advances in the field

**Reply:** *We would like to thank the reviewer for this comment. This was done. we have included the latest studies.*

2- The upper bound of the cumulative probability function $H(x)$ in Equation 21 is incorrectly defined to include the value 1. This leads to an undefined expression:

$$ln\left(\left(\frac{1}{1-H(x)}\right)^2\right)$$

when $H(x) = 1$, which evaluates to $ln(\infty)$. This is mathematically invalid and computationally dangerous, as it can cause overflow or undefined behavior in implementation. The formula must be constrained to $H(x) \in (0,1)$, not $[0,1]$. This critical issue must be addressed and corrected in the manuscript before further consideration.

**Reply:** *We would like to thank the reviewer for this comment. We replaced "≤" by "<".*

3- Even though MMK (Modified Mann-Kendall) is applied, the paper doesn't explain how the lag selection was determined for autocorrelation adjustments. Additionally, there are no ACF or PACF plots provided to support the chosen ARIMA model order.

**Reply:** *We would like to thank the reviewer for this comment. This was included and explained under the results section. The study utilised auto_arima() function instead of ACF or PACF.*

4-      Before using ARIMA, the study should have checked for stationarity using tests like ADF, but this step isn't mentioned. Also, it's unclear whether the data was normalized or scaled before being fed into the LSTM model or not.

**Reply:** *We would like to thank the reviewer for this comment. This was included and explained under the results section. The study utilised Box-Jenkins methodology to check the stationarity as the first step. On fitting the LSTM, the data normalization was applied and explained in the process of the hybrid model (see Figure 5).*

5- While the baseline models (ARIMA, LSTM, and CEEMDAN-LSTM) provide useful predictions, the study lacks a formal statistical comparison such as ANOVA to objectively assess their accuracy differences.

*Reply: We thank the reviewer for this important comment. We have noticed the following with the idea to use ANOVA:*

***Assumptions of ANOVA do not hold for time series Independence****: ANOVA assumes that observations are independent. In time series, values are autocorrelated (today's value depends on yesterday's), which violates this assumption.*

*Normality of residuals: Forecast errors in models like ARIMA or LSTM are not always normally distributed, especially in nonlinear settings.*

*Equal variance (homoscedasticity): Time series errors often have changing variance (heteroscedasticity).*

*Thus, because of these violations, the classical ANOVA test can give misleading results.*

*ANOVA is typically used to test if mean differences between groups are statistically significant.*

*In forecasting, the focus is not usually on comparing means but on predictive accuracy — how close forecasts are to observed values.*

*Metrics like RMSE, MAE, MAPE, NSE, $R^2$ are more informative than ANOVA F-tests.*

*We should mention that we did try it though, and it was apparent that Running an ANOVA on forecast errors assume residual independence, which is not valid because errors are serially correlated.*

*We also invite the reviewer to view any paper that does time series forecasting. All of them avoid using ANOVA, but rather use Metrics like RMSE, MAE, MAPE, NSE, $R^2$. In this paper we go one step further by using directional symmetry as well.*

*See example paper in this link:*
*https://scholar.google.com/scholar?hl=en&as_sdt=0%2C5&q=time+series+forecasting+using+ARIMA%2C+LSTM+...&btnG=*